# On Robust Streaming for Learning with Experts: Algorithms and Lower Bounds

**David P. Woodruff**
CMU
dwoodruf@cs.cmu.edu

**Fred Zhang**
UC Berkeley
z0@berkeley.edu

**Samson Zhou**
Texas A&M University
samsonzhou@gmail.com

## Abstract

In the online learning with experts problem, an algorithm makes predictions about an outcome on each of $T$ days, given a set of $n$ experts who make predictions on each day. The algorithm is given feedback on the outcomes of each day, including the cost of its prediction and the cost of the expert predictions, and the goal is to make a prediction with the minimum cost, compared to the best expert in hindsight. However, often the predictions made by experts or algorithms at some time influence future outcomes, so that the input is adaptively generated.

In this paper, we study robust algorithms for the experts problem under memory constraints. We first give a randomized algorithm that is robust to adaptive inputs that uses $\widetilde{O}\left(\frac{n}{R\sqrt{T}}\right)$ space for regret $R$ when the best expert makes $M = O\left(\frac{R^2 T}{\log^2 n}\right)$ mistakes, thereby showing a smooth space-regret trade-off. We then show a space lower bound of $\widetilde{\Omega}\left(\frac{nM}{RT}\right)$ for any randomized algorithm that achieves regret $R$ with probability $1 - 2^{-\Omega(T)}$. Such an algorithm is useful for adaptive inputs, as the failure probability is low enough to union bound over all computation paths. Our result implies that the natural deterministic algorithm, which iterates through pools of experts until each expert in the pool has erred, is optimal up to polylogarithmic factors. Finally, we empirically demonstrate the benefit of using robust procedures against a white-box adversary that has access to the internal state of the algorithm.

## 1 Introduction

*Online learning with experts* is a fundamental problem in sequential prediction. On each of $T$ days, an algorithm must make a prediction about an outcome, given a set of $n$ experts who make predictions on the outcome. The algorithm is then given feedback on the cost of its prediction and on the expert predictions for the current day. In the *discrete prediction with experts* problem, the set of possible predictions is restricted to a finite set, and the cost is 0 if the prediction is correct, and 1 otherwise. More generally, we assume the costs are restricted to be in a range $[0, \rho]$ for some fixed parameter $\rho > 0$, with lower costs indicating better performance. This process continues for the $T$ days, after which the performance (total cost) of the algorithm is compared to the performance (total cost) of the best performing expert. In particular, the goal for the online learning with experts problem is to minimize the regret, which is the amortized difference between the total cost of the algorithm and the total cost of the best performing expert, i.e., the expert that incurs the least overall cost.

A well-known folklore algorithm for handling the discrete prediction with experts problem is the weighted majority algorithm [42]. The deterministic variant of the weighted majority algorithm simply initializes "weights" for all experts to 1, down-weights any incorrect expert on a given day, and selects the prediction supported by the largest weight of experts. The algorithm solves the discrete prediction with experts problem with $O(M + \log n)$ total mistakes, where $M$ is the number of mistakes made by the best expert, thus achieving total regret $O(M + \log n)$. More generally, a large

body of literature has studied optimizations to the weighted majority algorithm, such as a randomized variant where the probability of the algorithm selecting each prediction is proportional to the sum of the weights of the experts supporting the prediction. The randomized weighted majority algorithm achieves regret $O\left(\sqrt{\log n/T}\right)$ [42], which has been shown to be information-theoretically optimal, up to a constant. There have subsequently been many follow-ups to the weighted and randomized weighted majority algorithms that achieve similar regret bounds, but improve in other areas. For example, on a variety of structured problems, such as online shortest paths, follow the perturbed leader [38] achieves the same regret bound as randomized weighted majority but uses less runtime on each day. In addition, the multiplicative weights algorithm achieves the optimal $\sqrt{\ln n/(2T)}$ regret, with a tight leading constant [33]. However, these classic algorithms use a framework that maintains the cumulative cost of each expert, which requires the algorithm to store $\Omega(n)$ bits of information.

**Memory bounds.** Recently, [49] considered the online learning with experts problem when memory is a premium for the algorithm. On the hardness side, they showed that any algorithm achieving a target average regret $R$ requires $\Omega\left(\frac{n}{R^2 T}\right)$ space, which implies that any algorithm achieving the information-theoretic $O\left(\sqrt{\log n/T}\right)$ regret must use near-linear space. On the other hand, when the number of mistakes $M$ made by the best expert is small, i.e., $M = O\left(R^2 T\right)$, [49] gave a randomized algorithm that uses $\widetilde{O}\left(\frac{n}{RT}\right)$ space for arbitrary-order streams, thus showing that the hardness of their lower bound originates from a setting where the best expert makes a large number of mistakes.

Subsequently, [47] considered the online learning with experts problem when the algorithm is limited to memory sublinear in $n$. They introduced a general framework that achieves $o(T)$ regret using $o(n)$ memory, with a trade-off parameter between space and regret that obtains $O_n\left(T^{4/5}\right)$ regret with $O\left(\sqrt{n}\right)$ space and $O_n\left(T^{0.67}\right)$ regret with $O\left(n^{0.99}\right)$ space.

**Adaptive inputs.** Up to now, the discussion has focused on an oblivious setting, where the input to the algorithm may be worst-case, but is chosen independently of the algorithm and its outputs. The online learning with experts problem is often considered in the adaptive setting, where the input to the algorithm is allowed to depend on previous outputs by the algorithm, e.g., in financial markets, future stock quotes can depend on previous investment choices. Formally, we define the adaptive setting as a two-player game between an algorithm $\mathcal{D}$ and an adversary $\mathcal{A}$ that adaptively creates the input stream to $\mathcal{D}$. The game then proceeds in days and on the $t$-th day:

(1) The adversary $\mathcal{A}$ chooses the outputs of all experts on day $t$ as well as the outcome of day $t$, depending on all previous stream updates and all previous outputs from the algorithm $\mathcal{D}$.

(2) The outputs (i.e., predictions) of all experts are simultaneously given to the algorithm $\mathcal{D}$, which updates its data structures, acquires a fresh batch $R_t$ of random bits, and outputs a predicted outcome for day $t$.

(3) The outcome of day $t$ is revealed to $\mathcal{D}$, while the predicted outcome for day $t$ by $\mathcal{D}$ is revealed to the adversary $\mathcal{A}$.

The goal of $\mathcal{A}$ is to induce $\mathcal{D}$ to make as many incorrect predictions as possible throughout the stream. It is clear that any deterministic algorithm for the online learning with experts problem will maintain the same guarantees in the adaptive model. Unfortunately, both the algorithms of [49] and [47] are randomized procedures that rely on iteratively sampling "pools" of experts, which can potentially be exploited by an adaptive adversary who learns the experts sampled in each pool. Interestingly, both the randomized weighted majority algorithm [42] and the multiplicative weights algorithm [33] are known to be robust to adaptive inputs.

## 1.1 Our Contributions

In this paper, we study the capabilities and limits of sublinear space algorithms for the online learning with experts problem on adaptive inputs.

**Robust algorithms.** Towards adaptive robustness, it is natural to study deterministic algorithms, since they retain the same guarantee under adaptive adversaries. As a warm-up, we first provide a simple deterministic algorithm that uses space $\widetilde{O}\left(\frac{nM}{RT}\right)$. Consider an algorithm that iteratively

selects the next pool of $k = \widetilde{O}\left(\frac{nM}{RT}\right)$ experts and running the deterministic majority algorithm on the experts in the pool, while removing any incorrect experts from the pool until the pool is completely depleted, at which point the next pool of $\widetilde{O}\left(\frac{nM}{RT}\right)$ experts is selected. The main intuition is that each pool can incur at most $O\left(\log n\right)$ mistakes before it is depleted and the best expert can only make $M$ mistakes. By the time the pool has cycled through $nM$ experts, i.e., $M$ times for each of the $n$ experts, then the best expert no longer makes any mistakes and will be retained by the pool. Thus, the total number of mistakes made by the algorithm is $\frac{nM}{k} \cdot O\left(\log n\right)$. On the other hand, for a target average regret $R$, the mistake bound of the algorithm is required to be at most $M + RT$, so it suffices to set $k = \widetilde{O}\left(\frac{nM}{RT}\right)$ to achieve regret $R$. Since the algorithm runs deterministic majority on a pool of $k = \widetilde{O}\left(\frac{nM}{RT}\right)$ experts, then this algorithm uses $\widetilde{O}\left(\frac{nM}{RT}\right)$ space. Formally, we show:

**Theorem 1.1** (Simple deterministic algorithm; see Section 3.1). *Suppose the best expert makes $M$ mistakes and let $R \geq \frac{4M \log n}{T}$. There exists a deterministic algorithm (Algorithm 2) that uses space $\widetilde{O}\left(\frac{nM}{RT}\right)$ and achieves an average regret of $R$.*

The algorithm is simple, computationally efficient, and easy to implement. However, the drawback is that for $M = \Omega(RT)$, the algorithm requires space near-linear in the number of experts $n$, which is undesirable when $n$ is large. To address this issue, we complement our deterministic algorithm with a randomized algorithm that is robust to adaptive inputs and allows for a different memory-regret trade-off:

**Theorem 1.2** (Robust randomized algorithm). *Let $R > \frac{64 \log^2 n}{T}$, and suppose the best expert makes at most $M \leq \frac{R^2 T}{128 \log^2 n}$ mistakes. Then there exists an algorithm for the discrete prediction with experts problem that uses $\widetilde{O}\left(\frac{n}{R\sqrt{T}}\right)$ space and achieves regret at most $R$, with high probability.*

This gives a trade-off between the space and regret, almost all the way to the information-theoretic limit of $R = O_n\left(\sqrt{1/T}\right)$ for general worst-case input. However, it incurs a multiplicative space overhead of $\widetilde{O}(\sqrt{T})$ compared to the optimal algorithms for oblivious input. Thus we believe the complete characterization of the space complexity of the discrete prediction with experts problem with adaptive input is a natural open question resulting from our work.

**Tight memory bounds for robust algorithms.** It is natural to ask whether there exist robust algorithms that are more space-efficient than the straightforward deterministic approach. For example, [12] showed that any oblivious randomized algorithm with failure probability $2^{-\Omega(nT)}$ will be robust against adaptive outputs in the discrete prediction with experts problem, so a reasonable approach would be to boost the success probability of existing oblivious algorithms to $1 - 2^{-\Omega(nT)}$. Unfortunately, we show this cannot work:

**Theorem 1.3** (Memory lower bound for high-probability algorithms). *For $n = o(2^T)$, any randomized algorithm algorithm that achieves $R$ regret with probability at least $1 - 2^{-\Omega(T)}$ for the discrete prediction with experts problem must use $\Omega\left(\frac{nM}{RT}\right)$ space when the best expert makes $M$ mistakes.*

In particular, Theorem 1.3 shows that any deterministic algorithm must use $\Omega\left(\frac{nM}{RT}\right)$ space, which taken together with the deterministic procedure above, resolves the deterministic streaming complexity of online learning with experts. We emphasize that Theorem 1.3 also shows that using the strategy of high-probability randomized algorithms to guarantee robustness against adaptive input does not work any better than a deterministic algorithm.

At a conceptual level, our lower bound in Theorem 1.3 shows that surprisingly, the number $M$ of the mistakes made by the best expert is an intrinsic parameter that governs the abilities and limitations of robust algorithms in this model. Thus, even though $M$ is not a parameter that may naturally be ascertained in practice, it nevertheless completely characterizes the complexity of the problem. On the other hand, for algorithmic purposes, it suffices to acquire a constant-factor approximation to $M$ as an input to the algorithm.

Another reason Theorem 1.3 is somewhat surprising is because as the number of mistakes $M$ made by the best expert increases, then the algorithm is also permitted to make more mistakes and in some sense, the problem seems "easier". However, Theorem 1.3 shows this intuition is not true—the problem actually becomes more difficult as $M$ increases.

Moreover, we give an alternative proof in the regime when $M = \Omega(T)$. The proof differs from the proof of Theorem 1.3. Instead, it leverages the communication complexity of a new set disjointness problem, recently proposed by [39]. The statement is technically weaker Theorem 1.3 and appears in the appendix; see Appendix E.

**Empirical evaluations.** Finally, we conduct experimental evaluations in Section 5 by comparing the natural deterministic algorithm to the randomized algorithm of [49] against a white-box adversary who has access to the internal state of the algorithm, including any experts sampled and maintained by the algorithm. The deterministic algorithm iteratively selects pools of $k = \widetilde{O}\left(\frac{nM}{RT}\right)$ experts, discarding any expert that has erred, and refreshing the pool with the next batch of $k$ experts once the pool is emptied. The randomized algorithm similarly discards erroneous experts from a pool of $k$ experts, but it repeatedly samples pools of $k$ experts rather than selecting the next pool of $k$ experts. On average across the multiple trials for each setting, the randomized algorithm made several times more mistakes than the deterministic algorithm, ranging from 1.98x times more mistakes to 3.29x times more mistakes than the deterministic algorithm, thus demonstrating the importance of robust algorithms against adversarial inputs.

## 1.2 Related Work

**The experts problem and memory bounds.** The experts problem has been extensively studied [17], both in the discrete decision setting [42] and in the setting where costs are determined by various loss functions [35, 52–55]. Hence, the experts problem can be applied to many different applications, such as portfolio optimization [24, 23], ensemble boosting [32], and forecasting [37]. Given certain assumptions on the expert, such as assuming the experts are decisions trees [36, 50], threshold functions [43], or have nice linear structures [38], additional optimizations have been made to improve the algorithmic runtimes for the experts problem and more generally, existing work has largely ignored optimizing for memory constraints in favor of focusing on time complexity or regret guarantees, thus frequently using $\Omega(n)$ memory to track the performance of each expert.

Recently, [49] introduced the study of memory-regret trade-offs for the experts problem. For $n \gg T$, [49] showed that the space complexity of the problem is $\tilde{\Theta}\left(\frac{n}{R^2 T}\right)$ in the random-order streams, but also gave a randomized algorithm that uses $\widetilde{O}\left(\frac{n}{RT}\right)$ space for arbitrary-order streams when the number of mistakes $M$ made by the best expert is "small". Subsequently, [47] considered the online learning with experts problem for $T \gg n$, introducing a general space-regret trade-off framework that achieves $o(T)$ regret using $o(n)$ memory, including $O_n(T^{4/5})$ regret with $O(\sqrt{n})$ space and $O_n(T^{0.67})$ regret with $O(n^{0.99})$ space.

**Concurrent and independent work.** Concurrent to our work, [46] considered a variant of the problem where at each time, the algorithm selects an expert instead of a prediction. They then introduce an algorithm robust against an adaptive adversary who observes the specific expert chosen by the algorithm at each time, as well as lower bounds for any algorithm robust to such an adversary.

One way to ensure adversarial robustness is through deterministic algorithms. On that end, we achieve stronger lower bounds for deterministic algorithms, showing that there must be a dependency on the number $M$ of mistakes made by the best expert, i.e., any deterministic algorithm achieving amortized regret $R$ must use $\widetilde{\Omega}\left(\frac{nM}{RT}\right)$ space. In fact, when the number of mistakes $M$ made by the best expert is sufficiently small, i.e., $M = O\left(\frac{R^2 T}{\log^2 n}\right)$ for amortized regret $R$, we give a randomized upper bound that uses *less* space than this lower bound. By comparison, the lower bound of [46] shows that any algorithm achieving $R$ amortized regret must use $\widetilde{\Omega}\left(\sqrt{\frac{n}{R}}\right)$ space, though their lower bound also applies to randomized algorithms.

Due to the difference in setting, our algorithmic techniques are quite different from those of [46]. We use a recent idea of [34, 4, 10] to hide the internal randomness of our algorithm from the adversary whereas [46] rotates between groups of experts to prevent an adversary from inducing high regret by making a specific expert bad immediately after it is selected.

## 2 Preliminaries

For any $t \leq n$ and vector $(X_1, X_2, \cdots, X_n)$, we let $X_{<t}$ denote $(X_1, \cdots, X_{t-1})$, $X_{\leq t} = (X_1, \cdots, X_t)$, and $X_{-t} = (X_1, \cdots, X_{t-1}, X_{t+1}, \cdots, X_n)$. Also, $X_{>t}$ and $X_{\geq t}$ are defined similarly. Let $e_i$ denote the $i$th standard basis vector, and for any $S$, $e_S$ the vector that has a 1 at index $i \in S$ and 0 everywhere else. For a random variable $X$, let $H(X)$ denote its entropy.

We write $[n]$ for an integer $n > 0$ to denote the set $\{1, \ldots, n\}$. We write $\mathrm{poly}(n)$ to denote a fixed polynomial in $n$. If an event occurs with probability at least $1 - \frac{1}{\mathrm{poly}(n,T)}$, we say the event occurs with high probability. We give additional technical preliminaries in Appendix B.

**Formal problem statement.** In the online learning with experts problem, there are $n$ experts that each make predictions on each of $T$ days. The prediction are in $\{0, 1\}$. An algorithm uses the experts to output a prediction for each day $t \in [T]$. The actual outcome of the day $t$ is then revealed, at which point the algorithm is penalized with a cost that is 0 if the prediction is correct, and 1 otherwise.

This process continues for the $T$ days. At the end, suppose that the best expert has incurred cost $M$, while the algorithm has incurred $C$. Then the performance of the algorithm is measured by the (average) regret $R = \max\left(\frac{C-M}{T}, 0\right)$.

**Differential privacy.** We use tools from differential privacy.

**Definition 2.1** (Differential privacy, [30]). *Given a privacy parameter $\varepsilon > 0$ and a failure parameter $\delta \in (0, 1)$, a randomized algorithm $\mathcal{A} : \mathcal{X}^* \to \mathcal{Y}$ is $(\varepsilon, \delta)$-differentially private if, for every pair of neighboring streams $S$ and $S'$ and for all $E \subseteq \mathcal{Y}$,*

$$\mathbf{Pr}\left[\mathcal{A}(S) \in E\right] \leq e^\varepsilon \cdot \mathbf{Pr}\left[\mathcal{A}(S') \in E\right] + \delta.$$

**Theorem 2.2** (Private median, e.g., [34]). *Given a database $\mathcal{D} \in X^*$, a privacy parameter $\varepsilon > 0$ and a failure parameter $\delta \in (0, 1)$, there exists an $(\varepsilon, 0)$-differentially private algorithm PRIVMED that outputs an element $x \in X$ such that with probability at least $1 - \delta$, there are at least $\frac{|S|}{2} - m$ elements in $S$ that are at least $x$, and at least $\frac{|S|}{2} - m$ elements in $S$ that are at most $x$, for $m = O\left(\frac{1}{\varepsilon} \log \frac{|X|}{\delta}\right)$.*

**Theorem 2.3** (Advanced composition, e.g., [31]). *Let $\varepsilon, \delta' \in (0, 1]$ and let $\delta \in [0, 1]$. Any mechanism that permits $k$ adaptive interactions with mechanisms that preserve $(\varepsilon, \delta)$-differential privacy guarantees $(\varepsilon', k\delta + \delta')$-differential privacy, where $\varepsilon' = \sqrt{2k \ln \frac{1}{\delta'}} \cdot \varepsilon + 2k\varepsilon^2$.*

**Theorem 2.4** (Generalization of DP, e.g., [29, 9]). *Let $\varepsilon \in (0, 1/3)$, $\delta \in (0, \varepsilon/4)$, and $n \geq \frac{1}{\varepsilon^2} \log \frac{2\varepsilon}{\delta}$. Suppose $\mathcal{A} : X^n \to 2^X$ is an $(\varepsilon, \delta)$-differentially private algorithm that curates a database of size $n$ and produces a function $h : X \to \{0, 1\}$. Suppose $\mathcal{D}$ is a distribution over $X$ and $S$ is a set of $n$ elements drawn independently and identically distributed from $\mathcal{D}$. Then*

$$\Pr_{S \sim \mathcal{D}, h \leftarrow \mathcal{A}(S)}\left[\left|\frac{1}{|S|}\sum_{x \in S} h(x) - \mathbb{E}_{x \sim \mathcal{D}}[h(x)]\right| \geq 10\varepsilon\right] < \frac{\delta}{\varepsilon}.$$

## 3 Algorithms Against Adaptive Adversaries

In this section, we show that there exists algorithms for the discrete prediction with experts problem that is robust to adaptive outputs.

### 3.1 A Near-Optimal Deterministic Algorithm

We first present a simple deterministic algorithm for arbitrary-order streams. The algorithm repeatedly selects pools of the next $k = \widetilde{O}\left(\frac{nM}{RT}\right)$ experts. While the pool is non-empty, the algorithm runs the deterministic majority algorithm on the algorithm and removes any incorrect experts from the pool. Once the pool is empty, the next $\widetilde{O}\left(\frac{nM}{RT}\right)$ experts are added to the pool, possibly cycling through all $n$ experts multiple times if necessary, where an expert can be added to the pool again even if it has been previously deleted from the pool. We give the formal algorithm and analysis in Appendix C.

**Theorem 3.1** (Determistic algorithm). *Among $n$ experts in a stream of length $T$, suppose the best expert makes $M$ mistakes and let $R \geq \frac{4M \log n}{T}$. There exists a deterministic algorithm (Algorithm 2) that uses space $\widetilde{O}\left(\frac{nM}{RT}\right)$ and achieves an average regret of $R$.*

In light of lower bound Theorem 1.3, it is evident that Theorem 3.1 is nearly optimal, up to polylogarithmic factors, for deterministic algorithms, which are automatically adversarially robust. On the other hand, it does not seem necessary that any adversarially robust algorithm must be deterministic. Indeed, we now give a randomized adversarially robust algorithm with better space guarantees.

## 3.2 A Randomized Robust Streaming Algorithm

We first recall the following randomized algorithm for arbitrary-order streams with oblivious input, i.e., non-adaptive input:

**Lemma 3.2** (Algorithm for oblivious inputs; [49]). *Let $R > \sqrt{\frac{128 \log^2 n}{T}}$, and suppose the best expert makes at most $M \leq \frac{R^2 T}{1280 \log^2 n}$ mistakes. Then there exists an algorithm DISCPRED for the discrete prediction with experts problem that uses $\widetilde{O}\left(\frac{n}{RT}\right)$ space and achieves regret at most $R$, with high probability, i.e., probability at least $1 - \frac{1}{\text{poly}(n,T)}$.*

The algorithm of Lemma 3.2 for constant probability proceeds by sampling pools of $k = \widetilde{O}\left(\frac{n}{RT}\right)$ experts and running majority vote on the pool, while iteratively deleting poorly performing experts until no experts remain in the pool, at which a new pool of $k$ experts is randomly sampled. The main intuition is that either the pool of experts will perform well and achieve low regret, or the pool will be continuously re-sampled until the best expert is sampled multiple times, after which point it will not be deleted from the pool. Unfortunately, it is not evident that this algorithm is robust to adaptive inputs because an adversary can potentially learn the experts in each sampled pool and force the experts to make mistakes only on days in which they are sampled by the algorithm. To boost the algorithm to high probability of success, we take the deterministic majority vote of $O(\log n)$ independent instances of the algorithm with constant success probability.

Towards adaptive robustness, we use differential privacy to hide the internal randomness of the algorithm, and in particular, the identity of the experts that are sampled by each pool. We first run $\widetilde{O}(\sqrt{T})$ copies of the algorithm and then output the private median of the $\widetilde{O}(\sqrt{T})$ copies, guaranteeing roughly $\left(\frac{1}{\widetilde{O}(\sqrt{T})}, 0\right)$-differential privacy because we use $\widetilde{O}(\sqrt{T})$ copies of the algorithm. Advanced composition, i.e., Theorem 2.3, then ensures $(O(1), 1/\text{poly}(n))$-differential privacy, so that correctness then follows from the generalization properties of DP, i.e., Theorem 2.4.

We give our algorithm in full in Algorithm 1.

---

**Algorithm 1** Randomized, robust streaming algorithm for the experts problem

**Input:** A stream of length $T$ with $n$ experts and a target regret $R$
**Output:** A sequence of predictions with regret $R$

1: Run $m = O\left(\sqrt{T} \log(nT)\right)$ independent instances of DISCPRED with regret $\frac{R}{4}$
2: Run PRIVMED on the $m$ instances with privacy parameter $\varepsilon = O\left(\frac{1}{\sqrt{T} \log(nT)}\right)$ and failure probability $\delta = \frac{1}{\text{poly}(n,T)}$
3: At each time $t \in [T]$, select the output of PRIVMED

---

Next, we show the correctness of our algorithm on adaptive inputs.

**Theorem 3.3** (Algorithm for adaptive inputs). *Let $R > \sqrt{\frac{2048 \log^2 n}{T}}$, and suppose the best expert makes at most $M \leq \frac{R^2 T}{1280 \log^2 n}$ mistakes. Then there exists an algorithm for the discrete prediction with experts problem that uses $\widetilde{O}\left(\frac{n}{R\sqrt{T}}\right)$ space and achieves regret at most $R$, with probability at least $1 - \frac{1}{\text{poly}(n,T)}$.*

*Proof.* Suppose we run $m = O\left(\sqrt{T}\log(nT)\right)$ independent instances of DISCPRED with regret $\frac{R}{4}$.
Note that for $R > \sqrt{\frac{2048\log^2 n}{T}}$, we have $\frac{R}{4} > \sqrt{\frac{128\log^2 n}{T}}$, which is a valid input to DISCPRED in Lemma 3.2. By Lemma 3.2, each instance succeeds on an arbitrary-order stream with probability at least $1 - 1/\text{poly}(n,T)$. By a union bound over the $m$ instances, all instances succeed with probability at least $1 - 1/\text{poly}(n,T)$. In particular, each instance has regret at most $R/4$, so that the total number of mistakes by each instance is at most $M + RT/4$. Thus, the total number of mistakes by all instances is at most $m\left(M + RT/4\right)$.

To consider an adaptive stream, observe that PRIVMED is called with privacy parameter $O\left(1/\sqrt{T}\log(nT)\right)$ and failure probability $1/\text{poly}(n,T)$. By Theorem 2.3, the mechanism permits $T$ adaptive interactions and guarantees privacy $O\left(1\right)$ with failure probability $1/\text{poly}(n,T)$. By Theorem 2.4, we have that with high probability, if the output of the algorithm is incorrect, then at least $m/3$ of the instances DISCPRED are also incorrect. Since the total number of mistakes by all instances is at most $m\left(M + RT/4\right)$, then the total number of mistakes by the algorithm is at most $3\left(M + RT/4\right) \leq M + RT$, since $M \leq \frac{R^2 T}{1280\log^2 n}$. Hence, the algorithm achieves $R$ regret with high probability.

By Lemma 3.2, each instance of DISCPRED uses $\widetilde{O}\left(\frac{n}{RT}\right)$ space. Since we use $m = O\left(\sqrt{T}\log(nT)\right)$ independent instances of DISCPRED, then the total space is $\widetilde{O}\left(\frac{n}{R\sqrt{T}}\right)$. $\qquad\square$

## 4 Lower Bound for Arbitrary-Order Streams

In this section, we provide a space lower bound for randomized algorithms with a high probability of success. Together with Theorem 1.1, the lower bound completely characterizes the complexity of deterministic algorithms for the online learning with experts problem. We restate Theorem 1.3, give a proof sketch and defer the full analysis to Appendix D.

**Theorem 4.1** (Memory lower bound for high-probability algorithms). *For $n = o(2^T)$, any randomized algorithm algorithm that achieves $R$ regret with probability at least $1 - 2^{-\Omega(T)}$ for the discrete prediction with experts problem must use $\Omega\left(\frac{nM}{RT}\right)$ space when the best expert makes $M$ mistakes.*

*Proof sketch of Theorem 4.1.* We consider the communication problem of $\varepsilon$-DIFFDIST. It combines $n$ instances of the distributed detection problem given by [14]. This was also used by the prior work of [49] to prove space lower bounds for expert learning in random-order stream.

Specifically, for fixed $T$, the $\varepsilon$-DIFFDIST problem with $\varepsilon = \frac{M}{T}$ consists of $T$ players, who each hold $n$ bits, indexed from 1 to $n$. The players must distinguish between:

(1) the NO case $\mathcal{D}_{\text{NO}}^{(n)}$, in which every bit for every player is drawn i.i.d. from a fair coin and

(2) the YES case $\mathcal{D}_{\text{YES}}^{(n)}$, in which an index $L \in [n]$ is selected arbitrarily and the $L$-th bit of each player is chosen i.i.d. from a Bernoulli distribution with parameter $\left(1 - \frac{M}{T}\right)$, while all other bits for every player are chosen i.i.d. from a fair coin.

At a high level, the proof proceeds in two steps:

(1) First, we show that the $\varepsilon$-DIFFDIST problem can be reduced to the expert prediction problem in the streaming setting.

(2) Second, we prove a communication complexity lower bound for $\varepsilon$-DIFFDIST against any protocol that succeeds with probability $1 - 2^{-\Theta(T)}$, which includes deterministic protocols.

The first step is straightforward. In the reduction, each player in an instance of $\varepsilon$-DIFFDIST corresponds to a day of the expert problem. The $n$ bit input held by each player correspond to the $n$ expert predictions of each day. Therefore, in the NO case, each expert is correct on roughly half of the days. In the YES case, there is a single expert $L \in [n]$ that is correct on roughly $1/2 + \delta$ of the days (for $\delta = 1/2 - M/T$), while all other experts randomly guess each day. Suppose that there is a streaming

algorithm for the expert prediction problem with average regret $\delta/2$. Then roughly speaking, in the YES case, the algorithm is correct approximately on $1/2 + \delta/2$ of the days, while in the NO case where every expert is randomly guessing, the algorithm is correct on less than $1/2 + \delta/2$ of the days. This distinguishes the YES and NO case and thus solves $\varepsilon$-DIFFDIST.

For the second step, we show that solving the $\varepsilon$-DIFFDIST problem with probability at least $1-2^{-\Theta(T)}$ requires $\Omega(nM)$ total communication. We give a sketch of the argument below.

Observe that if the input is viewed as a $T \times n$ matrix, then $\mathcal{D}_{\text{NO}}^{(n)}$ is a product distribution across columns that can be written as $\zeta^n$, where $\zeta$ is the distribution over a single column such that all entries of the column are i.i.d. Bernoulli with parameter $\frac{1}{2}$. We view $\mathcal{D}_{\text{NO}}^{(n)}$ as a hard distribution and applies an information complexity analysis. By a direct sum argument, it suffices to show that the single column problem, i.e., distinguishing between $\mathcal{D}_{\text{NO}}^{(1)}$ and $\mathcal{D}_{\text{YES}}^{(1)}$ (i.e., for $n = 1$), requires $\Omega(M)$ total communication.

Let $(C_1, C_2, \ldots, C_T)$ be a single column drawn from the hard distribution—namely, the NO case where each player holds one i.i.d. Bernoulli with parameter $1/2$. Let $A$ be a fixed protocol with success probability at least $1 - \exp(-\Theta(T))$. For all $i < T$, let $M_i$ denote the message sent from player $P_i$ to player $P_{i+1}$ and $M_{<i} = \{M_j : j < i\}$. Let $\Pi = \Pi(C_1, \cdots, C_T)$ be the communication transcript of $A$ given the input $(C_i)_{i=1}^T$. A standard information complexity argument [8] implies that the total communication is at least the *information cost*, defined as $I(C_1, \ldots, C_T; \Pi(C_1, \ldots, C_T))$, where $I(X, Y)$ denotes the mutual information between random variables $X$ and $Y$.

The key step now is to lower bound the information cost by $\Omega(M)$. The main ideas are the following. For any $i \in [T]$, we say that $(M_i, M_{<i})$ is *informative* for $i$ with respect to the input $C$ and the transcript $\Pi = (M_1, M_2, \ldots, M_T)$ if

$$|\Pr(C_i = 0 \mid M_i, M_{<i}) - \Pr(C_i = 1 \mid M_i, M_{<i})| \geq c \tag{4.1}$$

for some constant $c > 0$. Otherwise, we say that $M_i$ is uninformative. Informally, an informative message $M_i$ reveals sufficiently large information about $C_i$ so that the mutual information $I(M_i, C_i \mid M_{<i})$ would be large. Let $p_i$ be the probability that $(M_i, M_{<i})$ is informative. Intuitively, we need that $\sum_i p_i$ is large, because then there would be sufficiently many informative messages, and so the information cost is high. To formalize this approach, we claim two key lemmas. First, by Lemma D.9

$$I(\Pi; C_1, C_2, \ldots, C_T) = \sum_{j=1}^T I(M_j; C_j \mid M_{<j}) \geq \Omega\left(\sum_{j=1}^T p_j\right).$$

Conceptually, this shows that the information cost is at least the expected number of informative messages. Furthermore, by Lemma D.10, the latter is indeed high, and in particular, $\sum_j p_j \geq \Omega(M)$. Much of the technical work is dedicated to prove these lemmas. This finishes the proof since the communication complexity is lower bounded by the information cost. $\qquad\square$

## 5  Experimental Evaluations

In this section, we perform experimental evaluations as a simple proof-of-concept demonstrating the importance of deterministic algorithms against adversarial input.

**Experimental setup.**   We assume a white-box adversary with access to the internal state of the algorithm. We evaluate the natural deterministic algorithm that iteratively selects pools of $k = \widetilde{O}\left(\frac{nM}{RT}\right)$ experts, discarding any expert that has erred, and refreshing the pool with the next batch of $k$ experts once the pool is emptied. As a baseline, we compare to a randomized algorithm that repeatedly samples pools of $k = \widetilde{O}\left(\frac{nM}{RT}\right)$ experts, discarding any expert that has erred, and refreshing the pool with the next batch of $k$ sampled experts once the pool is emptied.

Provided that the best expert has not yet made $M$ mistakes, the adversary simply compels the experts in each pool to err. Once all experts have made at least $M$ mistakes, the adversary gives up and permits all subsequent predictions to be correct. It can be theoretically verified that against such an adversary, the deterministic algorithm is the optimal algorithm, in the sense that it achieves the smallest number of errors.

**Experimental details.** We first evaluate our experiments on the setting $n = 10000$, $M = 20$, and $T = 1000$ across various values of $R \in \{0.05, 0.1, 0.15, 0.2, 0.25, 0.3, 0.35\}$. For each setting of $R$, we ran the experiment 20 times, recording the runtime and number of errors by the algorithms in each repetition. We then computed the minimum, mean, and maximum number of errors by the randomized algorithm across all 20 repetitions. We then repeated the experimental setup for a $10x$ larger setting of $T$, i.e., $n = 10000$, $M = 20$, and $T = 1000$. Our experiments were performed on a 64-bit operating system using an AMD Ryzen 7 5700U CPU with 8.00 GB RAM and 8 cores with base clock 1.80 GHz.

**Results.** Our experiments show that the deterministic algorithm performs significantly better than the randomized algorithm. On average across the 20 trials for each setting, the randomized algorithm made several times more mistakes than the deterministic algorithm, ranging from 1.98x times more mistakes for the setting $n = 100000$, $M = 20$, $T = 1000$, $R = 0.05$ to 3.06x times more mistakes for the setting $n = 100000$, $M = 10$, $T = 10000$, $R = 0.3$. Even the best performance by a randomized algorithm over all trials, which occurred at the setting $n = 100000$, $M = 20$, $T = 1000$, $R = 0.05$, the randomized algorithm made 1.9x times more mistakes than the deterministic algorithm. Meanwhile, the worst performance by a randomized algorithm over all trials, which occurred at the setting $n = 100000$, $M = 10$, $T = 10000$, $R = 0.25$, the randomized algorithm made 3.29x times more mistakes than the deterministic algorithm. The average runtime was 98 seconds for each batch of 20 experiments for the setting of $n = 100000$, $M = 20$, $T = 1000$, $R = 0.05$ while the average runtime was 98 seconds for each batch of 20 experiments for the setting of $n = 100000$, $M = 10$, $T = 10000$, $R = 0.3$ was roughly 350 seconds. See Figure 1 for a summary.

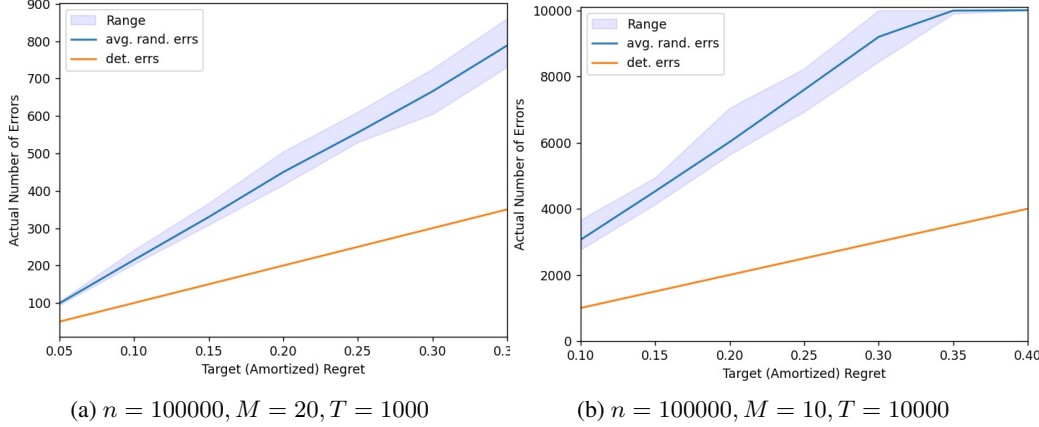

(a) $n = 100000$, $M = 20$, $T = 1000$      (b) $n = 100000$, $M = 10$, $T = 10000$

Figure 1: Comparison of errors made by deterministic algorithm and average number of errors made by randomized algorithm across 20 repetitions for each trial, across various values of input target regret $R$. Minimum and maximum numbers of errors by randomized algorithm across each trial are also reported.

## 6 Conclusion

In this work, we provide robust streaming algorithms for learning with experts. We provide a deterministic algorithm parametrized by the number of mistakes made by the best expert. We also give a randomized algorithm with a different space-regret trade-off, based on differential privacy. We complement our algorithms with a lower bound for high-probability success algorithms. This gives tight memory lower bound for deterministic algorithms. We then show the importance of robust algorithmic design by empirically comparing the performance of the natural deterministic algorithm and the state-of-the-art randomized algorithm when the inputs are adaptive.

We remark that our results do not rule out space-efficient robust algorithms that match the bounds of the oblivious randomized algorithm of [49] for constant probability of success. We believe whether or not there exists such an algorithm is a fascinating question for future work.

## Acknowledgments

David P. Woodruff and Samson Zhou were partially supported by a Simons Investigator Award and by the National Science Foundation under Grant No. CCF-1815840. This work was done in part while Fred Zhang was at Google Research, and Samson Zhou was at Carnegie Mellon University, UC Berkeley, and Rice University.

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

# A  Additional Related Work on Adaptive Inputs

Motivated by non-independent inputs and adversarial attacks, adaptive inputs have recently been considered in the centralized model [19, 41, 20, 21], in the streaming model [6, 12, 34, 57, 15, 18, 11, 1, 3, 22, 4, 28], and in the dynamic model [56, 10]. In particular, algorithms robust to inputs that can depend on the previous outputs by the algorithm, i.e., black-box attacks, are also robust to situations in which future inputs may be dependent on previous outputs. This is especially relevant in applications such as forecasting, in which a prediction on day $i$ can lead to a series of actions that might impact outcomes and expert predictions on day $i + 1$ and beyond.

Adaptive adversaries have received considerable attention in literature for online learning when the goal is simply to achieve the best possible regret [13, 17, 44]. Building off a line of results on multi-armed bandit problems [5, 7, 40], the work of [45] first considered the experts setting against memory-bounded adaptive adversaries, giving an algorithm with regret $O\left(T^{2/3}\right)$. An early paper of [25] introduced a family of algorithms for adaptive inputs, but provided guarantees using concepts not quite related to the standard definitions of regret. More recent works have explored online learning with additional considerations, such as alternative quantities to optimize [27], additional switching costs [16, 26, 48], and feedback graphs [2]. The closest work to our setting is the recent result by [47] showing that no algorithm using space sublinear in $n$ can achieve regret sublinear in $T$ when the input is chosen by an adversary with access to the internal state of the algorithm, i.e., a white-box adversary.

# B  Additional Technical Preliminaries

## B.1  Information Theory

For any $p \in [0, 1]$, we slightly abuse notation and let $H(p) = -p \log_2 p - (1 - p) \log_2 (1 - p)$ be the binary entropy function. The following is a standard upper and lower bound of $H(p)$.

**Lemma B.1** (Bound on the binary entropy function; see e.g. [51]). *For $p \in [0, 1]$, the binary entropy function satisfies*

$$4p(1 - p) \leq H(p) \leq 2(p(1 - p))^{1/\ln 4}.$$

## B.2  Communication Complexity

**Definition B.2** (Mutual information). *Let $X$ and $Y$ be a pair of random variables with joint distribution $p(x, y)$. Then the* mutual information *is defined as $I(X; Y) := \sum_{x,y} p(x, y) \log \frac{p(x,y)}{p(x)p(y)}$, for marginal distributions $p(x)$ and $p(y)$.*

In a multi-party communication problem of $t$ players, each player is given $x_i \in \mathcal{X}_t$. They communicate according to fixed protocol to compute a function $f : \mathcal{X}_t \times \cdots \times \mathcal{X}_t \to \mathcal{Y}$. A protocol $\Pi$ is called a $\delta$-error protocol for $f$ if there exists a function $\Pi_{\text{out}}$ such that $\Pr\left[\Pi_{\text{out}}\left(\Pi(x, y)\right) = f(x, y)\right] \geqslant 1 - \delta$. For a (multi-party) communication problem, we denote the transcript of all communication in a protocol as $\Pi \in \{0, 1\}^*$. The communication cost of a protocol, as a result, is the bit length of the transcript. Let $R_\delta(f)$ denote the minimum communication cost across all $\delta$-error protocols for $f$.

**Definition B.3** (Information cost). *Let $\Pi$ be a randomized protocol that produces a random variable $\Pi(X_1, \ldots, X_T)$ as a transcript on inputs $X_1, \ldots, X_T$ drawn from a distribution $\mu$. Then the information cost of $\Pi$ with respect to $\mu$ is defined as $I(X_1, \ldots, X_T; \Pi(X_1, \ldots, X_T))$.*

**Definition B.4** (Information complexity). *The information complexity of a function $f$ with respect to a distribution $\mu$ and failure probability $\delta$ is the minimum information cost of a protocol for $f$ with respect to $\mu$ that fails with probability at most $\delta$ on every input and denoted by $\mathsf{IC}_{\mu,\delta}(f)$.*

**Lemma B.5** (Information cost decomposition lemma, Lemma 5.1 in [8]). *Let $\mu$ be a mixture of product distributions and suppose $\Pi$ is a protocol for inputs $(X_1, \ldots, X_T) \sim \mu^n$. Then $I(X_1, \ldots, X_T; \Pi(X_1, \ldots, X_T)) \geq \sum_{i=1}^n I(X_{1,i}, \ldots, X_{T,i}; \Pi(X_1, \ldots, X_T))$, where $X_{i,j}$ denotes the $j$-th component of $X_i$.*

**Lemma B.6** (Information complexity lower bounds communication complexity; Proposition 4.3 [8]). *For any distribution $\mu$ and error $\delta$, $R_\delta(f) \geq \mathsf{IC}_{\mu,\delta}(f)$.*

# C   Proof of Theorem 3.1 and Formal Algorithm

We give a formal description of our deterministic robust algorithm in pseudocode.

---

**Algorithm 2** Deterministic algorithm for the experts problem

---

**Input:** A stream of length $T$ with $n$ experts, upper bound $M$ on the number of mistakes made by the best expert, and target regret $R$
**Output:** A sequence of predictions with regret $R$
1:  $k \leftarrow \frac{4nM}{RT} \log n$
2:  $S \leftarrow \emptyset$
3:  **while** the stream persists **do**
4:      **if** $S$ is empty **then**          ▷We have cycled through all $n$ experts once
5:          $S \leftarrow [n]$
6:      Let $P$ be the first $k$ indices of $S$
7:      $S \leftarrow S \setminus P$
8:      **while** $P \neq \emptyset$ **do**
9:          For each following day, choose the outcome output by the majority of the experts in $P$
10:         Delete the incorrect experts on that day from $P$

---

We now prove the correctness and space complexity of Algorithm 2.

*Proof of Theorem 3.1.*   We first remark that the algorithm can make at most $\log k \leq \log n$ mistakes over the lifespan of each pool of size $k := \frac{2nM}{RT} \log n$ because each time the algorithm makes a mistake, at least half of the pool must be incorrect and deleted, so the size of the pool decreases by at least half with each mistake the algorithm mistakes. Note that $k \leq n$ for $R \geq \frac{4M \log n}{T}$, so the algorithm is well-defined.

Since each pool $P$ has size $k$ and there are $n$ experts, then there are at most $\frac{4n}{k}$ pools before the entire set $S$, which is initialized to $n$, is depleted. Thus, there are at most $\frac{4n}{k}$ pools to iterate through the entire set of experts. Moreover, each time the algorithm has iterated through the entire set of experts, each expert must have made at least one mistake. This is because an expert is only deleted from the pool $P$ when it has made a mistake and since all experts have been deleted from $P$, then all experts have made at least one mistake.

Since the best expert makes at most $M$ mistakes, then the best expert can be deleted from the pool $P$ at most $M$ times. In other words, the algorithm can cycle through the entire set of $n$ experts at most $M + 1$ times.

Hence, the total number of mistakes by the algorithm is at most

$$\frac{2n}{k} \cdot \log n \cdot (M + 1) \leq \frac{4n}{k} \cdot \log n \cdot M \leq \frac{4nRT}{4nM \log n} \cdot \log n \cdot M = RT,$$

so the algorithm achieves regret at most $R$. Since the algorithm selects a subset of $k = \frac{4nM}{RT} \log n$ experts, then the space complexity follows.                    □

# D   Proofs of the Lower Bounds for Arbitrary-Order Streams

In this section, we give space lower bounds for the experts problem on arbitrary-order streams. As a warm-up, we first show in Section D.1 a general space lower bound for randomized algorithms when the best expert makes a "small" number of mistakes. We then give our main lower bound result in Section D.2, showing that any deterministic algorithm achieving regret $R$ must use space $\Omega\left(\frac{nM}{RT}\right)$ when the best expert makes $M$ mistakes.

## D.1   Warm-up: Lower Bound for Accurate Best Expert

In this section, we show that any randomized algorithm that achieves regret $R$ must use $\Omega\left(\frac{n}{RT}\right)$ space, even when the best expert makes $\Theta(RT)$ mistakes. In contrast, [49] give an $\Omega\left(\frac{n}{R^2T}\right)$ space lower bound:

**Theorem D.1** (Memory lower bound; Theorem 1 of [49]). *Let $R > 0$, $p < \frac{1}{2}$ be fixed constants, i.e., independent of other input parameters. Any algorithm that achieves $R$ regret for the experts problem with probability at least $1 - p$ must use at least $\Omega\left(\frac{n}{R^2 T}\right)$ space.*

*Furthermore, this lower bound holds even when the costs are binary, and expert predictions, as well as the correct answers, are constrained to be i.i.d. across the days, albeit with different distributions across the experts.*

The proof of this lower bound exploits a construction where the best expert makes $\Theta(T)$ mistakes. Thus, it is not clear how the space complexity of the problem behaves when the best expert makes a smaller number of mistakes. In fact, [49] also give an algorithm that uses $\widetilde{O}\left(\frac{n}{RT}\right)$ space when the best expert makes $O(RT)$ mistakes, bypassing the aforementioned lower bound.

We now prove that in this small mistake regime, this algorithm is tight. Towards this goal, we first define the $\varepsilon$-DIFFDIST problem that reduces to the experts problem. It was proposed by [49] to prove memory lower bounds for the expert problem in random order stream.

**Definition D.2** (The $\varepsilon$-DIFFDIST Problem). *We have $T$ players, each of whom holds $n$ bits, indexed from $1$ to $n$. We must distinguish between two cases, which we refer to as "$V = 0$" and "$V = 1$". Let $\mu_0$ be a Bernoulli distribution with parameter $\frac{1}{2}$, i.e., a fair coin, and let $\mu_1$ be a Bernoulli distribution with parameter $\frac{1}{2} + \varepsilon$.*

- *(NO Case, "$V = 0$") Every index for every player is drawn i.i.d. from a fair coin, i.e., $\mu_0$.*

- *(YES Case, "$V = 1$") An index $L \in [n]$ is selected arbitrarily—the $L$-th bit of each player is chosen i.i.d. from $\mu_1$. All other bits for every player are chosen i.i.d. from $\mu_0$.*

Any protocol that successfully solves the $\varepsilon$-DIFFDIST problem with a constant probability greater than $\frac{1}{2}$ must use at least $\Omega\left(\frac{n}{\varepsilon^2}\right)$ communication, a result due to [49]:

**Lemma D.3** (Communication complexity of $\varepsilon$-DIFFDIST; Lemma 3 of [49]). *The communication complexity of solving the $\varepsilon$-DIFFDIST problem with a constant $1 - p$ probability, for any $p \in [0, 0.5)$, is $\Omega\left(\frac{n}{\varepsilon^2}\right)$.*

The proof of Theorem D.1 by [49] uses $n$ coin flips across each of the $T$ players to form the $n$ expert predictions over each of the $T$ days. In the NO case, each expert will be correct on roughly $\frac{T}{2}$ days, while in the YES case, a single expert will be correct on roughly $\frac{T}{2} + \varepsilon T$ days, so that an algorithm with regret $R = O(\varepsilon)$ will be able to distinguish between the two cases. There is a slight subtlety in the proof that uses a masking argument to avoid "trivial" algorithms that happen to succeed on a "lucky" input, but for the purposes of our proof in this section, the masking argument is not needed. It then follows that the total communication is $\Omega\left(\frac{n}{R^2}\right)$ across the $T$ players, so that any streaming algorithm must use at least $\Omega\left(\frac{n}{R^2 T}\right)$ bits of space.

Suppose we instead consider the $\varepsilon$-DIFFDIST problem over $RT$ players, representing $RT$ days in the experts problem. Moreover, suppose we set $\varepsilon = \Theta(1)$ in the $\varepsilon$-DIFFDIST problem, so that in the NO case, each of the experts will be correct on roughly $\frac{RT}{2}$ days, while in the YES case, a single expert will be correct on roughly $\frac{RT}{2} + CRT$ days, for some constant $C > 0$. Suppose we further pad all of the experts with incorrect predictions across an additional $T - RT$ days, so that the total number of days is $T$, but the number of correct expert predictions remains the same. Then an algorithm achieving regret $O(R)$ will be able to distinguish between the two cases, so that the total communication is $\Omega\left(\frac{n}{R}\right)$, so that any streaming algorithm must use at least $\Omega\left(\frac{n}{RT}\right)$ bits of space.

**Corollary D.4.** *Let $R$, $p < \frac{1}{2}$ be fixed constants, i.e., independent of other input parameters. Any algorithm that achieves $R$ regret for the experts problem with probability at least $1 - p$ must use at least $\Omega\left(\frac{n}{RT}\right)$ space even when the best expert makes as few as $\Theta(RT)$ mistakes. This lower bound holds even when the costs are binary and expert predictions, as well as the correct answer, are constrained to be i.i.d. across the days, albeit with different distributions across the experts.*

*Proof.* The claim follows from setting $T = RT$ and $R = \Theta(1)$ in the proof of Theorem D.1. □

## D.2 Lower Bound for Deterministic Algorithms

We now prove our main space lower bound for deterministic algorithms (Theorem 1.3). We first set up some basic notations and introduce a hard distribution.

Let $T$ be any fixed positive integer. Let $\mathcal{D}_{\text{NO}}^{(n)}$ be the distribution over matrices $A$ with size $T \times n$ such that all entries of the matrix are i.i.d. Bernoulli with parameter $\frac{1}{2}$, i.e., each entry of $A$ is 0 with probability $\frac{1}{2}$ and 1 with probability $\frac{1}{2}$. Let $\mathcal{D}_{\text{YES}}^{(n)}$ be the distribution over matrices $M$ with size $T \times n$ such that there is a randomly chosen column $L \in [n]$, which is i.i.d. Bernoulli with parameter $\left(1 - \frac{M}{T}\right)$ and all other columns are i.i.d. Bernoulli with parameter $\frac{1}{2}$. Let $\text{BIASDETECT}_n$ be the problem of detecting whether $A$ is drawn from $\mathcal{D}_{\text{YES}}^{(n)}$ or $\mathcal{D}_{\text{NO}}^{(n)}$, so that $\text{BIASDETECT}_n$ is simply the $\varepsilon$-$\text{DIFFDIST}$ problem with $\varepsilon = \frac{1}{2} - \frac{M}{T}$.

Let $\Pi$ be a communication protocol for $\text{BIASDETECT}_n$ that is correct with probability at least $1 - \exp(-\Theta(T))$. Since $\mathcal{D}_{\text{NO}}^{(n)}$ is a product distribution across columns, then it can be written as $\zeta^n$, where $\zeta$ is the distribution over a single column such that all entries of the column are i.i.d. Bernoulli with parameter $\frac{1}{2}$. Let $\text{BIASDETECT}_1$ denote the problem of distinguishing between $\mathcal{D}_{\text{NO}}^{(1)}$ and $\mathcal{D}_{\text{YES}}^{(1)}$ on a single column, i.e., $n = 1$. Using $\mathcal{D}_{\text{NO}}^{(n)}$ as the hard distribution, we have the following direct sum theorem.

**Lemma D.5** (Direct sum for $\text{BIASDETECT}$). *The information complexity of $\text{BIASDETECT}_n$ satisfies*

$$\text{IC}_{\mathcal{D}_{\text{NO}}^{(n)}, 2^{-\Theta(T)}}(\text{BIASDETECT}_n) \geq n \cdot \text{IC}_{\mathcal{D}_{\text{NO}}^{(1)}, 2^{-\Theta(T)}}(\text{BIASDETECT}_1).$$

*Proof.* By definition, $\mathcal{D}_{\text{NO}}^{(n)} = \zeta^n$ is a product distribution over $n$ columns. The lemma follows from the standard direct sum lemma of information cost (Lemma B.5). $\square$

With the above direct sum theorem for $\text{BIASDETECT}_n$, it now suffices to provide a single-coordinate information cost lower bound against $\text{BIASDETECT}_1$. The proof is delayed to Section D.3.

**Lemma D.6** (Single-coordinate information cost lower bound). *Let $c \in (0, 1)$ and $\Pi$ be any protocol with error $\delta = 2^{-\Theta(T)}$ for $\text{BIASDETECT}_1$. We have that the information cost of $\Pi$ with respect to $\zeta$ is at least*

$$I(\Pi(C_1, C_2, \ldots, C_T); C_1, C_2, \ldots, C_T) \geq \Omega(M), \tag{D.1}$$

*where the bits $C_i \sim \zeta$ are i.i.d. single coordinates.*

Combining Lemma D.6 with the direct sum theorem (Lemma D.5), we immediately get the following information complexity lower bound for $\text{BIASDETECT}_n$:

**Theorem D.7** ($n$-Coordinate information complexity lower bound). *Let $c \in (0, 1)$. Then*

$$\text{IC}_{\mathcal{D}_{\text{NO}}^{(n)}, 2^{-\Theta(T)}}(\text{BIASDETECT}_n) = \Omega(nM).$$

*Proof.* This follows by applying the direct sum theorem (Lemma D.5) to the single-coordinate bound Lemma D.6. $\square$

This implies that any algorithm with $R$ regret and success rate at least $1 - 2^{-\Theta(T)}$ requires $\Omega\left(\frac{nM}{RT}\right)$ memory, where $M$ is the mistake bound on the best expert.

**Theorem D.8** (Memory lower bound for expert learning). *Let $R, M$ be fixed and independent of other input parameters. Any streaming algorithm that achieves $R$ regret for the experts problem with probability at least $1 - 2^{-\Theta(T)}$ must use at least $\Omega(\frac{nM}{RT})$ space, for $n = o\left(2^T\right)$, where the best expert makes $M$ mistakes.*

*Proof.* We now consider the problem $\text{BIASDETECT}_n$ on a matrix of size $RT \times n$. Note that in the NO case, at any fixed column $i \in [n]$, the probability that there are more than $\frac{3RT}{5} - \frac{M}{2}$ instances of 0, for $M \leq \frac{RT}{8}$, is at most $2\exp(-c_1 RT)$, for a sufficiently small constant $c_1 \in (0, 1)$. Thus, by a union bound, the probability that there exists an index $i \in [n]$ with more than $\frac{3RT}{4} - \frac{M}{2}$ instances of 0 is at most $2n\exp(-c_1 RT)$.

Similarly in the YES case, the probability that there are fewer than $\frac{4RT}{5} - \frac{M}{2}$ instances of 0 for a fixed $i \in [n]$ and for $M \leq \frac{RT}{8}$ is at most $2\exp(-c_2 RT)$, for a sufficiently small constant $c_2 \in (0, 1)$ and so by a union bound, the probability that there exists an index $i \in [n]$ with fewer than $\frac{3RT}{4} - \frac{M}{2}$ instances of 0 is at most $2n\exp(-c_2 RT)$. Hence, for $n = o(2^T)$, there exists a constant $c \in (0, 1)$ such that any algorithm that achieves total regret at most $\frac{RT}{5}$ with probability at least $1 - \exp(-cT)$ can distinguish between the YES and NO cases with probability $1 - \exp(-\Theta(T))$.

By Theorem D.7 and Lemma B.6, the total communication across the $RT$ players must be at least $\Omega(nM)$. Therefore, any streaming algorithm that achieves average $R$ regret for the experts problem with probability at least $1 - 2^{-\Theta(T)}$ must use at least $\Omega(\frac{nM}{RT})$ space. $\qquad\square$

### D.3 Proof of the Single-Coordinate Information Cost Lower Bound

We now show the single-coordinate lower bound of Lemma D.6.

*Proof of Lemma D.6.* Consider a protocol that is correct with probability $1 - 2^{-\Theta(T)}$ and let $(C_1, C_2, \ldots, C_T) \sim \zeta^T$ be a single column drawn from the NO case, where each coordinate is i.i.d. Bernoulli with parameter $1/2$. For notational convenience, let $\Pi = \Pi(C_1, \cdots, C_T)$ denote the transcript given the input $(C_1, C_2, \cdots, C_T)$. We consider the one-way message-passing model, where each player $P_i$ holds the input $C_i$. For all $i < T$, let $M_i$ denote the message sent from player $P_i$ to player $P_{i+1}$.

By the chain rule of mutual information, the information cost of the transcript, the left-side of Equation D.1 that we need to bound, can be written as

$$I(\Pi; C_1, C_2, \ldots, C_T) = \sum_{j=1}^{T} I\left(M_j; C_1, C_2, \ldots, C_T \mid M_{<j}\right). \tag{D.2}$$

By the independence of one-way communication, we have

$$I\left(M_j; C_1, C_2, \ldots, C_T \mid M_{<j}\right) = I\left(M_j; C_j \mid M_{<j}\right). \tag{D.3}$$

Combining the two equalities above, the information cost equals

$$I(\Pi; C_1, C_2, \ldots, C_T) = \sum_{j=1}^{T} I\left(M_j; C_j \mid M_{<j}\right). \tag{D.4}$$

We now lower bound the right-side. First, we make the following definition. For any $i \in [T]$, we say that $(M_i, M_{<i})$ is *informative* for $i$ with respect to the input $C$ and the transcript $\Pi = (M_1, \ldots, M_T)$ if

$$|\Pr(C_i = 0 \mid M_i, M_{<i}) - \Pr(C_i = 1 \mid M_i, M_{<i})| \geq c \tag{D.5}$$

for some constant $c > 0$; and uninformative otherwise. Intuitively, an informative index $i$ with respect to $(M_i, M_{<i})$ means that conditional on the past messages $M_{<i}$, the message $M_i$ reveals much information about $C_i$. Hence, in this case, $I(M_i, C_i \mid M_{<i})$ would be large. Now for all $i \in [T]$, let $p_i$ be the probability that $(M_i, M_{<i})$ is informative (for $i$ with respect to $C$ and $\Pi$).

Conceptually, we need to show that $\sum_i p_i$ is large, since then there would be sufficiently many informative messages, and so the information cost in the left-side of Equation D.4 is high. We formalize this idea in the following lemma.

**Lemma D.9.** *In the setting above, where $c > 0$ is a constant, the information cost can be lower bounded by*

$$I(\Pi; C_1, C_2, \ldots, C_T) = \sum_{j=1}^{T} I\left(M_j; C_j \mid M_{<j}\right) \geq \Omega\left(\sum_{j=1}^{T} p_j\right) \tag{D.6}$$

*Proof.* We start by expanding the definition of the mutual information terms. For each $j \in T$, we have

$$I\left(M_j; C_j \mid M_{<j}\right) = H\left(C_j \mid M_{<j}\right) - H\left(C_j \mid M_j, M_{<j}\right) \tag{D.7}$$

For the first term, notice that $C_j$ and $M_{<j}$ are independent by one-way communication. Moreover, by definition $C_j$ is Bernoulli with parameter $1/2$. Therefore,

$$H(C_j \mid M_{<j}) = H(C_j) = H(1/2) = 1.$$

For the second term,

- either $(M_j, M_{<j})$ is informative, which holds with probability $p_j$, and in this case, the conditional entropy is upper bounded by $H\left(C_j \mid M_j, M_{<j}\right) \leq H(1/2 + c/2)$;

- or $(M_j, M_{<j})$ is uninformative, and in this case, we trivially upper bound the conditional entropy by $H\left(C_j \mid M_j, M_{<j}\right) \leq 1$;

Putting the observations together and using Equation D.7, it follows that

$$\begin{aligned}
I\left(M_j; C_j \mid M_{<j}\right) &= H\left(C_j \mid M_{<j}\right) - H\left(C_j \mid M_j, M_{<j}\right) \\
&\geq 1 - \left(p_j \cdot H(1/2 + c/2) + (1 - p_j) \cdot 1\right) \\
&= p_j - p_j \cdot H(1/2 + c/2) \\
&\geq p_j \left(1 - (1 - c^2)^{1/\ln 4}\right).
\end{aligned}$$

where the last step uses the upper bound of Lemma B.1. Then we have

$$\begin{aligned}
I\left(M_j; C_j \mid M_{<j}\right) &\geq p_j \left(1 - (1 - c^2)^{1/\ln 4}\right) \\
&\geq c^3 \cdot \Omega(p_j),
\end{aligned}$$

where the last step follows since $1 - (1 - x^2)^{1/\ln 4} \geq x^3/100$ for $x \in [0, 1]$. Summing over $j = 1, 2, \ldots, T$ in Equation D.6 finishes the proof. $\square$

To prove the claimed information cost inequality Equation D.1, we show that $\sum_i p_i = \Omega(M)$.

**Lemma D.10.** *There exists a constant $\gamma > 0$ such that*

$$\sum_{j=1}^{T} p_j > \gamma \cdot M.$$

*Proof.* Suppose by way of contradiction that $\sum_{j=1}^{T} p_j = o(M)$. Let $A$ be a protocol that sends (possibly random) messages $M_1, \ldots, M_T$ on a random input $C \in \{0, 1\}^T \sim \zeta^T$ drawn uniformly from the NO distribution, i.e., each coordinate of $C := C_1, \ldots, C_T$ is picked to be 0 with probability $\frac{1}{2}$ and 1 with probability $\frac{1}{2}$. Moreover, suppose $A$ is a protocol that distinguishes between a YES instance and a NO instance with probability at least $1 - \frac{e^{-cT}2^{-T}}{8}$, for some constant $c > 0$.

Since $p_i$ is the probability that $M_i$ is informative, then by assumption, the expected number of informative indices $i$ over the messages $M_1, \ldots, M_T$ is $f(M)$ for some $f(M) = o(M)$. Thus by Markov's inequality, the probability that the number of informative indices is at most $10f(M) = o(M)$ with probability at least $\frac{9}{10}$. Let $S$ be the set of the uninformative indices so that $|S| = T - 10f(M) = T - o(M)$. Let $C'$ be an input that agrees with $C$ on the informative indices $[T] \setminus S$ and is chosen arbitrarily on uninformative indices $S$, so that $C'_i = C_i$ for $i \in [T] \setminus S$.

By definition, each uninformative index only changes the distribution of the output by a $(1 \pm c)$ factor. In particular, for $c \in (0, 1/2)$, the probability that the protocol $A$ generates $\Pi$ on input $C'$ is at least $(1 - c)^T \geq e^{-2cT}$ times the probability that the protocol $A$ generates $\Pi$ on input $C$. However, since $C$ can differ from $C'$ on $S$, then $C$ can differ from $C'$ on $|S| = T - 10f(M) = T - o(M)$ indices.

Now since each coordinate of $C$ is picked to be 0 with probability $\frac{1}{2}$ and 1 with probability $\frac{1}{2}$, then the probability that $C$ contains more than $T - M$ zeros is at least $1 - T^M \cdot \frac{1}{2^T} \geq 1 - 2^{T/2}$ for sufficiently large $T$. But then there exists a choice of $C'$ that contains fewer than $\frac{M}{2}$ zeros such that $A$ will also output $\Pi$ with probability at least $\frac{e^{-cT}}{2}$. Since $C'$ contains fewer than $\frac{M}{2}$, then $C'$ is more likely to generated from a YES instance and indeed a YES instance will generate $C$ with probability

$2^{-T}$. On the other hand, since $\Pi$ corresponds to a transcript for which $A$ will output NO, then the probability that $A$ is incorrect on $C'$ is at least $\frac{e^{-cT}}{4}$, which contradicts the claim that $A$ succeeds with probability $1 - \frac{e^{-cT}2^{-T}}{8}$. Thus it follows that $\sum_{j=1}^{T} p_j = \Omega(M)$, as desired. $\qquad\square$

Now we combine Lemma D.9 and Lemma D.10. This implies that the information cost can be lower bounded by

$$I(\Pi; C_1, C_2, \ldots, C_T) \geq \Omega\left(\sum_{j=1}^{T} p_j\right) \geq \gamma M, \tag{D.8}$$

for a constant $\gamma > 0$. This completes the proof. $\qquad\square$

# E  An Alternative Proof in the Large Mistake Regime

We give another analysis of the information cost when $M = \Omega(T)$, where $M$ is the number of mistakes of the best expert.

**Lemma E.1** (Single-Coordinate Information Cost Lower Bound). *Let $c \in (0,1)$ and $\Pi$ be any protocol with error $\delta = 2^{-T}$ for $\text{BIASDETECT}_1$. Suppose that the best expert makes $M = c'T$ mistakes for some constant $c'$. We have that the information cost of $\Pi$ with respect to $\zeta$ is at least*

$$I(\Pi(C_1, \cdots, C_T); C_1, \cdots, C_T) \geq \Omega\left((1-c)^2 T\right), \tag{E.1}$$

*where $C_i \sim \zeta$ are i.i.d. single coordinates.*

Applying direct sum theorem (Lemma D.5), we get the following information complexity lower bound for $\text{BIASDETECT}_n$:

**Theorem E.2** ($n$-Coordinate Information Complexity Lower Bound). *Let $c \in (0,1)$ and assume $M = c'T$ for some constant $c'$. Then*

$$\text{IC}_{\mathcal{D}^{(1)}, 2^{-\Theta(T)}}(\text{BIASDETECT}_n) = \Omega\left((1-c)^2 nT\right).$$

By an argument similar to Theorem D.8, we have:

**Theorem E.3** (Memory lower bound for expert learning). *Let $M = c'T$ for some constant $c'$. Any streaming algorithm that achieves constant regret for the experts problem with probability at least $1 - 2^{-\Theta(T)}$ must use at least $\Omega(n)$ space, where the best expert makes $M$ mistakes.*

For the purpose of proving Lemma E.1, we need some technical lemmas.

**Lemma E.4** (Lemma 3.5 of [39]). *Consider any communication protocol $\Pi$ where each player receives one bit and condition on any fixed input $b \in \{0,1\}^T$. Each player $i$ can be implemented such that, if the other players receive input $b_{-i}$, player $i$ only observes their input with probability $d_{TV}(\Pi_b, \Pi_{b \oplus e_i})$.*

**Lemma E.5** (Lemma 3.6 of [39]). *Let $c \in (0,1)$, $p \in (0, \frac{1-c}{2})$ and $\gamma_c = \frac{1}{c \log(e/c)}$. For a set of binary random variables $Y_1, Y_2, \cdots, Y_k$ such that $\mathbb{E}\left[\sum_i Y_i\right] = pk$, there exists a set $S \subset [n]$ of size $ck$ such that $\Pr(Y_j = 0, \forall j \in S) > e^{-k/\gamma_c - 1}$.*

*Proof of Lemma E.1.* Let $(C_1, C_2, \cdots, C_n) \sim \zeta^n$ be a single column drawn from the NO case, where each coordinate is i.i.d. Bernoulli with parameter $1/2$. Let $M = c'T$ for some constant $c'$. We consider the one-way message-passing model, where for all $i < T$, $M_i$ denotes the message sent from player $P_i$ to player $P_{i+1}$. It suffices to lower bound

$$I(\Pi; C_1, \cdots, C_T) = \sum_{j=1}^{T} I(\Pi; C_j | C_{<j}).$$

by the chain rule of mutual information. We claim that for any $j$

$$I(\Pi; C_j | C_{<j}) = I(\Pi; C_j | C_{-j}).$$

First, by data processing and the one-way nature of the protocol

$$I(\Pi; C_j | C_{<j}) = I(M_{\leq j}; C_j | C_{<j}).$$

for any $j$. Now we just need to show that

$$I(M_{\leq j}; C_j | C_{<j}) = I(\Pi; C_j | C_{-j}).$$

By chain rule of mutual information, we can write the right-hand side as

$$I(\Pi; C_j | C_{-j}) = I(M_{\leq j}; C_j | C_{-j}) + I(M_{>j}; C_j | M_{\leq j}, C_{-j})$$
$$= I(M_{\leq j}; C_j | C_{-j}) + I(M_{>j}; C_j | M_{\leq j}, C_{>j})$$

Observe that $M_{>j}$ and $C_j$ are independent, conditional on $M_{\leq j}$ and $C_{>j}$. Hence,

$$I(M_{>j}; C_j | M_{\leq j}, C_{>j}) = 0$$

and this proves the claim.

Let $\Pi_b$ be the distribution of the protocol transcript when the input is fixed to be $b \in \{0,1\}^n$ and $\oplus$ denote the binary XOR. Now we can bound

$$I(\Pi; C_1, \cdots, C_T) = \sum_{j=1}^{T} I(\Pi; C_j | C_{<j})$$

$$= \sum_{j=1}^{T} I(\Pi; C_j | C_{-j})$$

$$\geq \frac{1}{8} \frac{1}{2^T} \sum_{b \in \{0,1\}^T} \sum_{j=1}^{T} d_{\text{TV}}^2(\Pi_{b \oplus e_j}, \Pi_b)$$

$$\geq \frac{1}{8} \frac{1}{2^T} \sum_{b \in \{0,1\}^T} \sum_{j:b_j=0} d_{\text{TV}}^2(\Pi_{b \oplus e_j}, \Pi_b). \tag{E.2}$$

Conditioned on an input $b \in \{0,1\}^T$, let $k = |\{i : b_i = 0\}|$ and assume for the sake of a contradiction that

$$\sum_{i:b_i=0} d_{\text{TV}}(\Pi_{b \oplus e_i}, \Pi_b) = kp, \tag{E.3}$$

where $p < \frac{1-c}{2}$. Let $p_i = d_{\text{TV}}(\Pi_{b \oplus e_i}, \Pi_b)$ for every player $i \in [T]$. Lemma E.4 implies that the protocol can be equivalently implemented such that if the other players receive $b_{-i}$, player $i$ only looks at their input with probability $p_i$. If the player $i$ does not look at their bit, then their message $M_i$ is independent of their input bit. Let $Y_i$ denote the indicator random variable for the event that player $i$ looks at their input in this equivalent protocol.

It follows from our assumption (E.3) that if the input is $b$, then $\mathbb{E}\left[\sum_{i:b_i=0} Y_i\right] = \sum_i p_i = kp$. By the definition of $Y_i$, if for any set $S$, $Y_i = 0$ for all $i \in S$, then all players in $S$ do not look at their input bits. Let $E_S$ denotes the event that $Y_i = 0$ for all $i \in S$, for some $S \subseteq \{i : b_i = 0\}$. Then since the players in $S$ do not look at their input bits,

$$d_{\text{TV}}(\Pi_{b \oplus e_S} | E_S, \Pi_b | E_S) = 0.$$

In particular, using this and the law of total probability, we get that

$$d_{\text{TV}}(\Pi_{b \oplus e_S}, \Pi_b) = \Pr(E_S) \cdot d_{\text{TV}}(\Pi_{b \oplus e_S} | E_S, \Pi_b | E_S) + \Pr(\overline{E_S}) \cdot d_{\text{TV}}(\Pi_{b \oplus e_S} | \overline{E_S}, \Pi_b | \overline{E_S})$$
$$\leq \Pr(\overline{E_S}). \tag{E.4}$$

By our assumption, $\mathbb{E}\left[\sum_{i:b_i=0} Y_i\right] = kp$ for $p < \frac{1-c}{2}$. Applying Lemma E.5, we obtain that there exists a set $S \subseteq \{i : b_i = 0\}$ with $|S| = ck$ such that $\Pr(E_S) \geq e^{-k/\gamma_c - 1}$. For any $k < (T-2)\gamma_c < T - 2$, we have $\Pr(E_S) > e\delta$, and so $\Pr(\overline{E_S}) < 1 - e\delta$. By Eqn. (E.4), $d_{\text{TV}}(\Pi_{b \oplus e_S}, \Pi_b) < 1 - e\delta$. Observe that $b \oplus e_S$ differs from $b$ by having $|S| = ck$ more 1's; and they have same value at all other coordinates. Recall that in a typical single-coordinate YES instance, there are $T - M$ number of 1's, which is $T/2 - M$ more than a typical NO instance. Now suppose this

gap $T/2 - M < ck$; then solving $\text{BIASDETECT}_1$ is at most as hard as distinguishing $b$ and $b \oplus e_S$. Hence, if we choose $c'$ such that $M = c'T > T/2 - ck$, then the protocol $\Pi$ fails with probability greater than $\delta$. This is a contradiction.

Thus, for any $b$ such that $ck = c \cdot |\{i : b_i = 0\}| > T/2 - M$,

$$\sum_{i:b_i=0} d_{\text{TV}}(\Pi_{b \oplus e_i}, \Pi_b) \geq \Omega\left(\frac{(1-c)T}{2}\right).$$

From (E.2) and Jensen's inequality,

$$I(\Pi; C_1, \cdots, C_T) \geq \Omega\left((1-c)^2 T\right).$$

This finishes the proof. □

