# OpenReview forum: "On Robust Streaming for Learning with Experts: Algorithms and Lower Bounds"
_NeurIPS.cc/2023/Conference — NeurIPS 2023 poster_

### Official Review · Reviewer_757a · 2023-06-16

**Soundness:** 2 fair
**Presentation:** 2 fair
**Contribution:** 2 fair
**Rating:** 5
**Confidence:** 2

**Summary:**

This paper studied the memory complexity of the expert learning problem against an *adaptive* adversary. It proved a lower memory bound for deterministic algorithm and gave an optimal deterministic algorithm matching this bound. It also proposed a randomized algorithm which uses less memory if the regret and best expert performance satisfies certain condition.

**Strengths:**

1. The paper studies an interesting and important question. It seems to be the first work to study the memory requirement against an adaptive adversary for learning from expert problem specifically.
2. The paper has some interesting ideas, like the construction of the randomized algorithm and the proof of the lower bound, however, both of the writing and the results could be improved.

**Weaknesses:**

1. I believe your theorems are correct. However, some parts of your proof doesn't seem rigorous to me or could just be wrong. For example, in line 272, you claim "at least $m/3$ of the instance". I don't know where the constant $1/3$ comes from because you seem to be quite sloppy with the constants here (big O notation).



2. The writing should be improved drastically. \
(1). There are some typos. For example, in line 36, the regret should be divided by $T$. In line 206, "in $S$" is typed twice. \
(2). The proof needs to be cleaned up. There are lots of repetitions and unnecessary parts. For example, the theorems and definitions of differential privacy in the main article is very confusing to me. I don't think you used anything related to differential privacy either in your proof. Instead, you could just apply Theorem 1.4 in Hassidim et. al.. In appendix, some theorems proved in other papers are reproved. Instead, you should've highlighted the difference. Many of your proofs are just rewriting what's in Ben-Eliezer et. al. and Srinivas's paper. \
(3). Some notations are not so consistent. For example, high probability is defined by poly in $n$, but in your theorems, it's poly in both $n$ and $T$. In line 25, you mentioned the range of cost function is $[0,\rho]$ but in the problem you study is just $\{0,1\}$. \
(4). Many sentences are too long. Try to limit every sentence to less than three commas.



3. I think stronger result should be obtained. All of your current results just come from directly applying a theorem for more general settings. For example, the randomized algorithm directly comes from the framework in Hassidim et. al.. It seems that by designing algorithm more specific to the expert learning problem, better bound could be achieved. Your proof for the deterministic lower bound also seems quite unnecessary. I think a much simpler proof, potentially without using randomness, should suffice to prove the lower bound for deterministic algorithms. Or I think better bound (indeed the current lower bound doesn't really tell much about randomized algorithms because the exponential dependent on $T$ in the probability), like the probability is poly in $T$ and $n$, should be obtained by the current proof method. In the proof of the lower bound, you should highlight what's new. It seems that a large part like the reduction has already been showed by previous work.



4. You should at least compare the best randomized algorithm you derived, i.e., algorithm 1, with the deterministic algorithm in your experiments. Also why does the adversary have access to the internal randomness? Even the adaptive adversary doesn't have directly access to the random bits.




**Questions:**

I suggest the authors strengthen their results and rewrite the paper.

**Limitations:**

I think the results of the papers are somewhat exaggerated. For example, in the abstract the authors claimed the natural deterministic algorithm is almost optimal. I think this is quite misleading because better randomized algorithm could exist.

---

> ### Author Rebuttal · Authors · 2023-08-09
>
> > I believe your theorems are correct. However, some parts of your proof doesn't seem rigorous to me or could just be wrong. For example, in line 272, you claim "at least $m/3$ of the instance". I don't know where the constant $1/3$ comes from because you seem to be quite sloppy with the constants here (big O notation).
>
> Thanks for the feedback, we will elaborate that with high probability, at least $m/3$ of the $m$ instances of the algorithm must be wrong if the private median of the $m$ independent instances is wrong. This is because the private median will likely select an output in between the $1/3$ and $2/3$ quantiles among the outputs of all the instances.
>
> > The writing should be improved drastically.
>
> Algorithm 1 and  Theorem 3.3 builds upon the basic definition of private median, which in turns relies on a set of minimal preliminaries of differential privacy (DP). Otherwise, our algorithm's description would not be well-defined. We believe DP as a tool for algorithm design is not standard within the ML community, and thus we believe a basic introduction to it is necessary for our purpose.
>
> Note that our lower bound works under the special case when all losses are binary. This implies the same lower bound holds when loss values are continuous.  Our randomized robust algorithm works for general costs $[0,\rho]$ by using Algorithm 5 of Srinivas, Woodruff, Xu & Zhou (in line 1 of Algorithm 1 of our paper).
>
> We have fixed the mechanical errors and would be happy to  continue improving, if you may provide further suggestions.
>
> > I think stronger result should be obtained. All of your current results just come from directly applying a theorem for more general settings. For example, the randomized algorithm directly comes from the framework in Hassidim et. al.. It seems that by designing algorithm more specific to the expert learning problem, better bound could be achieved. Your proof for the deterministic lower bound also seems quite unnecessary. I think a much simpler proof, potentially without using randomness, should suffice to prove the lower bound for deterministic algorithms. Or I think better bound (indeed the current lower bound doesn't really tell much about randomized algorithms because the exponential dependent on $T$ in the probability), like the probability is poly in $T$ and $n$, should be obtained by the current proof method. In the proof of the lower bound, you should highlight what's new. It seems that a large part like the reduction has already been showed by previous work.
>
> We agree that our randomized algorithm is straightforward from existing techniques from DP; however, prior to our work,  there was no known  connection from adaptive robust via DP to  online learning.
>
> We are certain, on the other hand, that most of our lower bound techniques are novel, and we strongly disagree with the above statement of the reviewer regarding our main lower bound and do not see any evidence for the reviewer's claim. In particular, our core information-complexity argument, starting with the definition of "informativeness" (Eqn 4.1 and below) and using a non-trivial input modification technique depending on the protocol's success probability, does not follow from any existing techniques in the literature, e.g., in communication complexity, information complexity, or streaming lower bounds. We believe this technique may be of independent interest, and apply to proving deterministic (or high probability) lower bounds for other communication problems.
>
> We do not know how to strenthen it to a inverse-polynomial failure rate. We are, of course, happy to hear more constructive suggestions towards this goal.
>
> > You should at least compare the best randomized algorithm you derived, i.e., algorithm 1, with the deterministic algorithm in your experiments. Also why does the adversary have access to the internal randomness? Even the adaptive adversary doesn't have directly access to the random bits.
>
> Our randomized algorithm relies upon the prior work of SWXZ22. We are not aware of an implementation of the algorithm therein, as it is somewhat sophisticated. We leave a more comprehensive experimental study to future work, as the focus of this paper is mostly theoretical.
>
> Assuming the adversary has access to the internal randomness is a setting known as the white-box adversarial model (proposed by https://dl.acm.org/doi/pdf/10.1145/3517804.3526228). Our experimental evaluations serve as a proof-of-concept that our deterministic algorithms can be useful in certain settings.

---

> > ### Comment · Reviewer_757a · 2023-08-10
> >
> > Thank you very much for your response. I admit I underestimated the significance of your lower bound and the novelty of your proof techniques. I will adjust my rating and confidence score accordingly. However, I expect to see an improvement in your writing, with a main focus on explaining the lower bound.

---

> > > ### Author Response · Authors · 2023-08-15
> > >
> > > Thank you for the update! The NeurIPS conference setup does not allow for uploading a revised draft, unfortunately. The final version of our paper will address your comments. Meanwhile, would you be able to adjust your rating at this moment?

---

> > > > ### Comment · Reviewer_757a · 2023-08-15
> > > >
> > > > I've already updated my rating.

---

### Official Review · Reviewer_BvCn · 2023-07-04

**Soundness:** 4 excellent
**Presentation:** 4 excellent
**Contribution:** 2 fair
**Rating:** 6
**Confidence:** 3

**Summary:**

The paper studies the experts problem in online learning under an additional memory constraint. Recent development on this setup gave algorithms with a sublinear memory usage (in $n$, the number of experts) and also a sublinear regret (in $T$, the time horizon). However, many of these results require the observtions to be oblivious, i.e., not adaptively chosen against the learner's actions.

This work gives new bounds on the memory-regret tradeoff when the adversary is adaptive, including (under certain parameter regimes):
- A simple, deterministic algorithm that requires $\tilde O\left(\frac{nM}{RT}\right)$ space, where $M$ is the number of mistakes made by the best expert and $R$ is the desired regret. This is achieved by running the deterministic majority algorithm on repeatedly sample pools of experts.
- A randomized algorithm that uses $\tilde O\left(\frac{n}{R\sqrt{T}}\right)$ space. This builds on an algorithm of Srinivas, Woodruff, Xu and Zhou (STOC'22) (for the oblivious case) and uses tools from differential privacy to hide the random coins used by parallel copies of an algorithm.
- To achieve regret $R$ with failure probability $2^{-\Omega(T)}$, $\Omega\left(\frac{nM}{RT}\right)$ memory is needed. In particular, this lower bound holds against all deterministic algorithms. The lower bound is proved by a reduction from a communication problem, which was also used in a very similar context by Srinivas, Woodruff, Xu and Zhou (STOC'22).

**Strengths:**

The paper presents progress on a natural and fundamental problem in sequential prediction. The paper is very nicely structured and enjoyable to read. The main paper contains a nice amount of technical ideas/details to understand the work.

Despite the weaknesses discussed below, this work still seems a nice addition to the recent line of work on memory bounds for the experts problem, so I lean towards acceptance.

**Weaknesses:**

- The results are restricted to the discrete prediction setup (i.e., with binary outcomes, binary forecasts, and 0/1 loss), whereas some previous work along this line applies to the more general bounded-loss case. (Lines 25-26 mentions a setup with losses bounded in $[0, \rho]$, but all the results of the paper are stated for the discrete case.)
- Both the positive results (Theorems 3.1 and 3.3) require $RT \gg M$ (at least by a $\mathrm{polylog}(n)$ factor), i.e., the additional mistakes made by the algorithm needs to be much larger than that of the best expert.
- The algorithms proposed in the paper are relatively simple (or simple modification to algorithms in the literature).

**Questions:**

Regarding the second point in the "Weaknesses" section: Are there impossibility results showing the hardness of achieving regret $R \ll M/T$?

Minor comment:
- Abstract, Line 10: It was not clear what $M$ is here (before reading through Line 13).

**Limitations:**

The main limitations are the assumptions on the problem parameters in order for the theoretical results to hold, and these have been clearly stated in the paper.

---

> ### Author Rebuttal · Authors · 2023-08-09
>
> > The results are restricted to the discrete prediction setup (i.e., with binary outcomes, binary forecasts, and 0/1 loss), whereas some previous work along this line applies to the more general bounded-loss case. (Lines 25-26 mentions a setup with losses bounded in $[0,\rho]$, but all the results of the paper are stated for the discrete case.)
>
> Note that our lower bound works under the special case when all losses are binary. This implies the same lower bound holds when loss values are continuous. Our randomized robust algorithm works for general costs $[0,\rho]$ by using Algorithm 5 of Srinivas, Woodruff, Xu & Zhou (in line 1 of Algorithm 1 of our paper).
>
> > Both the positive results (Theorems 3.1 and 3.3) require $RT\gg M$ (at least by a $\text{polylog}(n)$ factor), i.e., the additional mistakes made by the algorithm needs to be much larger than that of the best expert.
>
> Our main result shows that the regime $RT\gg M$ is absolutely necessary. In other words, our lower bound in Theorem 1.3 shows that if $R\ll M/T$, then any deterministic algorithm (or even randomized algorithm with failure probability $2^{-\Omega(T)}$) requires linear space.
>
> > The algorithms proposed in the paper are relatively simple (or simple modification to algorithms in the literature).
>
> We remark that we view our main result as the lower bound, which shows that even though the number of mistakes by the best expert is difficult to ascertain a priori, it nevertheless is a hidden parameter that governs the space complexity of the problem. Our proof requires non-trivial input modification techniques surrounding our notion of informative indices.
>
> > Regarding the second point in the "Weaknesses" section: Are there impossibility results showing the hardness of achieving regret $R\ll M/T$?
>
> Yes, our main result in Theorem 1.3 shows that if $R\ll M/T$ or in other words $RT\ll M$, then any deterministic algorithm (or even randomized algorithm with failure probability $2^{-\Omega(T)}$) requires $\Omega(n)$ space, i.e., essentially keeping all the experts, up to a constant factor.

---

> > ### Comment · Reviewer_BvCn · 2023-08-14
> > **Thank you for your reply!**
> >
> > I want to thank the authors for answering my questions, especially for clarifying the necessity of $RT \gg M$ in the positive results. Accordingly, I will change the overall evaluation to "weak accept". Still, I would encourage the authors to update the writing to: (1) include the discussion on general costs; (2) highlight the necessity of $RT \gg M$; (3) emphasize that the lower bound is the main contribution.

---

### Official Review · Reviewer_Tz12 · 2023-07-04

**Soundness:** 3 good
**Presentation:** 3 good
**Contribution:** 3 good
**Rating:** 6
**Confidence:** 2

**Summary:**

 This paper considers the problem of individual sequence prediction with a finite pool of experts, under memory constraints. While this problem was considered in previous works when the learner faces oblivious adversaries. The authors consider the case of adaptive adversaries that choose the experts costs in each round $t$ based on past information. They present two algorithms, the first algorithm is deterministic and the second is randomized. Both algorithms are adaptive the cumulative loss of the best expert. Finally, the authors provide space lower-bounds for any randomized algorithm, which along with the upper bound of their deterministic algorithm characterizes the space-regret trade-off for this problem.

**Strengths:**

 The adaptability to cases where the best-performing expert has small losses and when the adversary takes the learner's previous decisions into account are desirable features in practice. The presented algorithms are intuitive and well explained

**Weaknesses:**

* The algorithm requires prior knowledge of the number of mistakes by the best expert and a pre-specified target regret.
* In line 25 you announce that you are considering losses in $[0,\rho]$, while you only consider $\{0,1\}$ losses.

**Questions:**

* In the non-constrained setting, a refined analysis of the regret of the Hedge algorithm with an adequate parameter allows to have a regret of order $\mathcal{O}\left(\sqrt{M\log(n)/T}\right)$ using the papers' notation (result corresponds to Corollary 2.4 in [1]). How do the upper bounds presented in this paper apply to algorithms with the last guarantees?

[1] Cesa-Bianchi, N., & Lugosi, G. (2006). Prediction, learning, and games. Cambridge university press.

**Limitations:**

---

> ### Author Rebuttal · Authors · 2023-08-09
>
> > The algorithm requires prior knowledge of the number of mistakes by the best expert and a pre-specified target regret.
>
> We remark that the natural deterministic algorithm only requires a loose upper bound on the number of mistakes by the best expert. Moreover, it is not the focus of our paper. Our main result is the lower bound, which involves non-trivial input modification techniques surrounding our definition of informative indices, and shows that even though the number of mistakes made by the best expert is difficult to ascertain a priori, it nevertheless is a hidden parameter that governs the space complexity of the problem.
>
> > In line 25 you announce that you are considering losses in $[0,\rho]$, while you only consider $0,1$ losses.
>
> Note that our lower bound works under the special case when all losses are binary. This implies the same lower bound holds when loss values are continuous.  Our randomized robust algorithm works for general costs $[0,\rho]$ by using Algorithm 5 of Srinivas, Woodruff, Xu & Zhou (in line 1 of Algorithm 1 of our paper).
>
> > In the non-constrained setting, a refined analysis of the regret of the Hedge algorithm with an adequate parameter allows to have a regret of order $\mathcal{O}(\sqrt{M\log(n)/T})$ using the papers' notation (result corresponds to Corollary 2.4 in [1]). How do the upper bounds presented in this paper apply to algorithms with the last guarantees?
>
> Hedge works in the regime of memory size $\Theta(n)$. In this setting, our determinstic algorithm (which works for online binary prediction) reduces to a simple algorithm due to Littlestone & Warmuth (see e.g., Fact 11.3 here: https://www.cs.cmu.edu/~anupamg/advalgos17/scribes/lec11.pdf). We get $M\log n$ mistakes, up to a constant.  On the other hand, our randomized algorithm has total regret at least $\sqrt T$ (up to logarithmic factors), due to the assumption that $R > \sqrt{\frac{\log^2 n}{T}}$, where $R$ is the average regret. This can be achieved when we have $\Omega(n)$ memory; hence, our randomized algorithm is not better than Hedge in this regime.

---

> > ### Comment · Reviewer_Tz12 · 2023-08-14
> >
> > Thanks for the reply. I don't have further questions and will keep my score as is for now.

---

### Official Review · Reviewer_gy2q · 2023-07-06

**Soundness:** 3 good
**Presentation:** 2 fair
**Contribution:** 3 good
**Rating:** 6
**Confidence:** 3

**Summary:**

The paper studies the learning with experts problem with a focus on: 1) the trade-off between the memory usage and the regret; 2) robustness against an adaptive adversary who decides the experts’ predictions and the correct prediction on each time step, based on all previous stream updates and the player’s outputs. Below are the three main results:
- A simple deterministic algorithm (thus robust to adaptive inputs) that iteratively run the majority vote algorithm on subsets of the experts of size $\tilde O(\frac{nM}{RT})$ where $n$ is the number of experts, $M$ the number of mistakes of the best expert, $T$ the number of time steps, and $R$ the regret target of the algorithm. The algorithm therefore uses $\tilde O(\frac{nM}{RT})$ space to achieve regret $R$.
- A randomized algorithm that is robust against adaptive input, uses only $\tilde O(\frac{n}{R\sqrt T})$ space, and has regret at most $R$, with probability at least $1-\frac{1}{poly(n,T)}$. The algorithm runs copies of the random, memory-efficient learning with experts algorithm from [49] (it’s unclear if this algorithm is robust to adaptivity). Differential privacy is used to hide the randomness against the adaptive adversary.
- A memory lower bound of $\Omega(\frac{nM}{RT})$ for any algorithm which achieves regret R with probability at least $1-2^{-\Omega(T)})$ against an oblivious (i.e. not adaptive) adversary. In particular, this gives an $\Omega(\frac{nM}{RT})$ memory lower bound for all deterministic algorithms, showing that their simple deterministic algorithm is in fact optimal.


**Strengths:**

The deterministic algorithm proposed is simple and achieves the optimal memory usage $O(nM/RT)$. In addition, it’s surprising to see that $M$, the number of mistakes made by the best expert, shows up both in the lower and upper bound of memory, implying that $M$ is an intrinsic parameter related to the robust variant of learning with expert problem. This might provide insights for other related problems.

The randomized algorithm shows differential privacy can be used to turn (potentially) non-robust algorithms to robust ones by hiding the internal randomness. This idea was explored in recent works [4, 10, 34], and it’s interesting to see its application for the expert problem.


**Weaknesses:**

Most techniques in the proofs are already used in previous works. For example, the proof of the space lower bound is by a reduction to the communication problem of \epsilon-DIFFDIST. Similar reduction is already used in the closely related work [49] to show memory lower bound for learning with experts problem with random-order stream.

The memory lower bound only seems to rule out using oblivious algorithms for adaptive adversaries. In particular, this does not prevent the existence of some algorithm that would have a standard inverse polynomial probability of failure for adaptive adversaries. The lower bound also asks the number of experts to be exponentially large in $T$ ($n\geq 2^T$), which is also restrictive. It could be useful if the authors could expand on whether this is a reasonable regime for lower bounds.

The algorithms need to know $M$, the number of mistakes of the best expert, which in practice, one cannot expect to know a priori. Although a constant approximation is enough, it seems unclear whether this can lead to an oblivious algorithm in $M$.

In the experiment, the adversary uses a “greedy” strategy by compelling the experts in the pool that the algorithm uses to make mistakes (until all experts have made at least M mistakes). It would be nice to compare the deterministic and the random algorithms against more types of strategies for the adversary. (As already pointed out in the paper, for such “greedy” adversary, the deterministic algorithm is theoretically optimal.)


**Questions:**

- The paper proposes that the difficulty of the problem might depend on M, the number of mistakes made by the best experts, and that the space required increases as M increases. In addition, the guarantee for the random algorithm proposed holds only for small M. In the classical learning with experts problem, similar phenomenon is observed. For instance, for small M the regret is in fact $O(\frac{1}{T}(\sqrt{M\log n}+\log n)) (section 2.4 in [Cesa-Bianchi, N., & Lugosi, G. (2006)]). Is any connection between the classical setting and the variant considered here, in terms of the effect of small M?

Some minor issues:
- Line10 $R$ has not been introduced yet. It is also unclear what $M$ is at this point.
- Line 36, the $O(M+\log n)$ term seems to be the “total” regret not the average regret as in the rest of the paper, and it’s a bit confusing by saying it’s the regret.
- line 73, the goal of the adversary stated here is a bit misleading. If my understanding is correct, the guarantees in the theorems are for the regret, not the number of mistakes made by the algorithm.
- line 119 "algorithm algorithm" typo
- the main theorems are stated twice (in section 1 and section 3), which is not necessary especially given the tight 9-page constraint
- The notations big-O, O_n, and \tilde{O} are not clearly defined. It could confusing which terms are suppressed.


References:

Cesa-Bianchi, N., & Lugosi, G. (2006). Prediction, Learning, and Games. Cambridge: Cambridge University Press. doi:10.1017/CBO9780511546921


**Limitations:**

Limitations have been addressed in the paper.

---

> ### Author Rebuttal · Authors · 2023-08-09
>
> > Most techniques in the proofs are already used in previous works. For example, the proof of the space lower bound is by a reduction to the communication problem of $\epsilon$-DIFFDIST. Similar reduction is already used in the closely related work [49] to show memory lower bound for learning with experts problem with random-order stream.
>
> We remark that although the communication problem itself is the same as [49], our lower bound statement and the corresponding techniques are completely different from [49], in order to handle a stronger lower bound for deterministic algorithms, parameterized by the number of mistakes made by the best expert.
>
> At a high level, the proof of our main lower bound proceeds in two steps. First, we prove a communication complexity lower bound for  $\epsilon$-DIFFDIST against any protocol that succeeds with probability $1-2^{-\Theta(T)}$, which includes deterministic protocols. Second, we show that the $\epsilon$-DIFFDIST problem can be reduced to the expert prediction problem in the streaming setting. Indeed, the second step was done previously by [49], but the first step forms the major technical contribution of our paper.
>
> Namely, we first view the input as a $T\times n$ matrix and define a hard distribution that is a product distribution across columns that can be written as $\zeta^n$, where $\zeta$ is the distribution over a single column. By a direct sum argument, it suffices to show that the single column problem, i.e., distingushing $\zeta$ from some other input distribution $\zeta'$, requires $\Omega(M)$ information cost and thus $\Omega(M)$ total communication, where $M$ is the number of mistakes made by the best expert. We then define an informative message to be a message that reveals sufficiently large information about the input so that the mutual information would be large.
>
> We show that there must exist $\Omega(M)$ informative indices because otherwise we could modify the input $C$ drawn from $\zeta$ on the uninformative indices and find an input $C'$ on which the algorithm cannot guarantee correctness with probability at least $1-\exp(-\Theta(T))$. Namely, we let $C'$ be an input that agrees with $C$ on the informative indices, and is chosen arbitrarily on uninformative indices. Thus, there must exist $\Omega(M)$ informative indices and by the chain rule for mutual information, the total communication must be $\Omega(M)$.
>
> We emphasize these techniques are specifically catered for our lower bound against deterministic algorithms and thus have not been used in previous works. We believe our technique of modifying the input to agree on informative indices may be useful for other problems for proving lower bounds for deterministic protocols.
>
> > The memory lower bound only seems to rule out using oblivious algorithms for adaptive adversaries. In particular, this does not prevent the existence of some algorithm that would have a standard inverse polynomial probability of failure for adaptive adversaries.
>
> Our lower bound is against algorithms with exponentially low failure rate, including all **deterministic** algorithms. We believe this is a wide space for robust algorithm design that contains highly non-trivial candidates, making our results fairly strong. For example, the classic algorithm for the online mistake bound model is the deterministic weighted majority scheme, due to Littlestone and Warmuth (1994). The algorithm, though simple, is arguably not obvious.
>
> > The lower bound also asks the number of experts to be exponentially large in $T$ ($n\ge 2^T$), which is also restrictive. It could be useful if the authors could expand on whether this is a reasonable regime for lower bounds.
>
> We emphasize that our lower bound asks the number of experts to be sub-exponential in $T$, i.e., $n=o(2^T)$ rather than $n\ge 2^T$. This is a **crucial difference** to what the reviewer is stating, and is actually a very **natural regime** of parameters.
>
> > In the experiment, the adversary uses a “greedy” strategy by compelling the experts in the pool that the algorithm uses to make mistakes (until all experts have made at least M mistakes). It would be nice to compare the deterministic and the random algorithms against more types of strategies for the adversary. (As already pointed out in the paper, for such “greedy” adversary, the deterministic algorithm is theoretically optimal.)
>
> Our experiments serve as a simple proof-of-existence of an adversary for which previous randomized algorithms fail catastrophically, demonstrating the need for an adversarially robust algorithm and the advantages of deterministic algorithms.
>
> > The paper proposes that the difficulty of the problem might depend on M, the number of mistakes made by the best experts, and that the space required increases as M increases. In addition, the guarantee for the random algorithm proposed holds only for small M. In the classical learning with experts problem, similar phenomenon is observed. For instance, for small M the regret is in fact $O(\frac{1}{T}(\sqrt{M\log n}+\log n)) (section 2.4 in [Cesa-Bianchi, N., & Lugosi, G. (2006)]). Is any connection between the classical setting and the variant considered here, in terms of the effect of small M?
>
> Thank you for pointing this out! Yes, we are aware of this coincidence, but not of a formal connection.

---

> > ### Comment · Reviewer_gy2q · 2023-08-14
> >
> > Thank you very much for your answers and comments. I do not have further questions and will keep my score.

---

### Author Rebuttal · Authors · 2023-08-09

We thank the reviewers for their thoughtful comments and valuable feedback. In particular, our experiments show the importance of algorithms that are robust to adversarial input, demonstrating that the existing algorithms perform poorly against a simple attack while deterministic algorithms are a natural class of adversarially robust algorithms. On the other hand, we noticed that multiple reviewers focused on the simplicity of the algorithmic design. We would therefore like to refocus the discussion on the stronger lower bound for deterministic algorithms (and algorithms with failure probability $2^{-O(n)}$) presented in this paper, which we view as the main contribution of our paper and exhibits several noteworthy properties:
- Our lower bound requires a novel and technical argument to show that protocols with an exponentially small probability of failure for even a single column of an input from a hard distribution of the $\epsilon$-DIFFDIST problem must convey $\Omega(M)$ information, where $M$ is the number of mistakes made by the best expert.
- Our lower bound shows that surprisingly, the number of mistakes made by the best expert is an intrinsic parameter that governs the complexity of the problem, regardless of whether it is known (or even approximated/bounded) a priori.
- Our lower bound also shows that the natural deterministic algorithm is the optimal algorithm, thus resolving this general approach.
 so it is important that our results completely resolve this avenue of attack.

We also remark that we appreciate the positive remarks, such as

- The deterministic algorithm proposed is simple and achieves the optimal memory usage $O(nM/RT)$. (Reviewer gy2q)
- In addition, it’s surprising to see that $M$, the number of mistakes made by the best expert, shows up both in the lower and upper bound of memory, implying that $M$ is an intrinsic parameter related to the robust variant of learning with expert problem. This might provide insights for other related problems. (Reviewer gy2q)
- The randomized algorithm shows differential privacy can be used to turn (potentially) non-robust algorithms to robust ones by hiding the internal randomness. This idea was explored in recent works [4, 10, 34], and it’s interesting to see its application for the expert problem. (Reviewer gy2q)
- The adaptability to cases where the best-performing expert has small losses and when the adversary takes the learner's previous decisions into account are desirable features in practice. (Reviewer Tz12)
- The presented algorithms are intuitive and well explained. (Reviewer Tz12)
- The paper presents progress on a natural and fundamental problem in sequential prediction. (Reviewer BvCn)
- The paper is very nicely structured and enjoyable to read. (Reviewer BvCn)
- The main paper contains a nice amount of technical ideas/details to understand the work. (Reviewer BvCn)
- This work still seems a nice addition to the recent line of work on memory bounds for the experts problem. (Reviewer BvCn)
- The paper studies an interesting and important question. (Reviewer 757a)
- It seems to be the first work to study the memory requirement against an adaptive adversary for learning from expert problem specifically. (Reviewer 757a)
- The paper has some interesting ideas, like the construction of the randomized algorithm and the proof of the lower bound. (Reviewer 757a)

We provide our responses to the initial comments of each reviewer below.  We hope our answers resolve all initial questions and concerns raised by the reviewers and we will be most happy to answer any remaining questions during the discussion phase!

---

### Decision · Program_Chairs · 2023-09-21

**Decision:**

Accept (poster)

**Comment:**

This is a clean paper that studies robust algorithms for the "learning with experts" problem under memory constraints. Robustness here means that the algorithm's input can be adaptively generated, depending on the experts' predictions in previous rounds. The paper pins the tradeoff between space complexity and regret for deterministic algorithms, but leaves the question for randomized algorithms open. The reviewers' questions were successfully addressed by the authors during the rebuttal phase.